# A Harmonized Global Land Evaporation Dataset from Model-based Products Covering 1980-2017

Jiao Lu[1], Guojie Wang[1]*, Tiexi Chen[1], Shijie Li[1], Daniel Fiifi Tawia Hagan[1], Giri Kattel[1,2,3], Jian Peng[4,5], Tong Jiang[1], Buda Su[1]

[1]Collaborative Innovation Center on Forecast and Evaluation of Meteorological Disasters, School of Geographical Sciences, Nanjing University of Information Science & Technology, Nanjing 210044, China
[2]Department of Infrastructure Engineering, University of Melbourne, Melbourne, VIC 3010, Australia
[3]Department of Hydraulic Engineering, Tshinghua University, Beijing, 100084, China
[4]Department of Remote Sensing, Helmholtz Centre for Environmental Research - UFZ, Permoserstrasse 15, 04318, Leipzig, Germany
[5]Remote Sensing Centre for Earth System Research, Leipzig University, Talstr. 35, 04103, Leipzig, Germany

*Correspondence to*: Guojie Wang (gwang@nuist.edu.cn)

**Abstract.** Land evaporation (ET) plays a crucial role in hydrological and energy cycle. However, the widely used model-based products, even though are helpful, are still subject to great uncertainties due to imperfect model parameterizations and forcing data. The lack of available observed data has further complicated the estimation. Hence, there is an urgency to define the global proxy land ET with lower uncertainties for climate-induced hydrology and energy change. This study has combined three existing model-based products: the fifth-generation ECMWF Re-Analysis (ERA5), Global Land Data Assimilation System Version 2 (GLDAS2) and the second Modern-Era Retrospective analysis for Research and Applications (MERRA2) to obtain a single framework of a long-term (1980-2017) daily ET product at a spatial resolution of 0.25°. Here, we use the Reliability Ensemble Averaging (REA) method, which minimizes errors to a reference data, to combine the three products over regions with high consistencies between the products using the coefficient of variation (CV). Global Land Evaporation Amsterdam Model Version 3.2a (GLEAM3.2a) and flux tower observation data were selected as the data for reference and evaluation, respectively. The results showed that the merged product performed well over a range of vegetation cover scenarios. The merged product also captured the trend of land evaporation over different areas well, showing the significant decreasing trend in Amazon plain in South America and Congo Basin in central Africa, and the increasing trend in the east of North America, west of Europe, south of Asia and north of Oceania. In addition to demonstrating a good performance, the REA method also successfully converged the models based on the reliability of the inputs. The resulting REA data can be accessed at https://doi.org/10.5281/zenodo.4595941 (Lu et al., 2021).

## 1 Introduction

Land evaporation plays an important role in the exchange of energy, water and carbon in the terrestrial biosphere, hydrosphere and atmosphere. It is one of the dominant components of land water and energy budget, as well as a key driver

of drought episode (Seneviratne, 2012; Sheffield et al., 2012). Therefore, it is important to quantify the spatial and temporal patterns of land evaporation. In addition, it is used to estimate water requirements for irrigation by agricultural and water resource management groups. The strength of the hydrological cycle determines water availability and affects the climate system in a variety of ways (Mueller et al., 2013). Apart from hydrological applications, land evaporation change is also related to air temperature change and extreme high temperature condition (Seneviratne et al., 2006, 2010; Hirschi et al., 2011; Mueller and Seneviratne, 2012). It is apparent that land evaporation is regarded as the intermediate variable of soil moisture affecting air temperature. Thus, it could be inferred that the uncertainty of land evaporation estimation will introduce adverse errors in various aspects, which creates the need for a global proxy ET data set with lower uncertainties.

Since the land surface is more heterogeneous than the ocean, it is difficult to estimate land evaporation accurately due to huge uncertainties resulting from complex land-atmosphere feedback processes. In addition, a major challenge remains to be addressed, for instance, there is no direct signal describing land evaporation that has been remotely detected. Recently, solar-induced chlorophyll fluorescence (SIF) has been discovered as an emerging technique to observe the photosynthetic processes of vegetation by quantifying the emission of fluorescent radiation (Joiner et al., 2014). Remotely sensed SIF has potential to empirically track the variation of canopy-level transpiration (Lu et al., 2018; Shan et al., 2019). However, satellite observations relating to surface temperature, soil moisture or vegetation coverage data can be combined with traditional flux formulas as an alternative method to derive global estimates at different temporal and spatial scales (Monteith, 1965; Priestley and Taylor, 1972). The available terrestrial ET datasets have widely varying estimates and even opposite long-term trends, indicating the existence of non-negligible uncertainties. Further, satellite-based land evaporation products show great discrepancies when compared with latent heat flux from flux towers (Jimenez et al., 2011; Mueller et al., 2011; McCabe et al., 2016; Peng et al., 2016, 2020). Although the latent heat flux is widely used as the benchmark for assessing the quality of land ET data sets, flux tower data is unevenly distributed around the world, only densely in some regions of North America, Europe and Oceania (as shown in Fig. 1a). On account of these reasons, an alternative product is developed which leverages the strengths of widely used existing model-based ET products.

In previous studies, far from hindering the use of these land evaporation data sets, differences among the products are capable of facilitating researches for the best merging methods to obtain data sets with lower uncertainties (Jimenez et al., 2018). Due to differences in the algorithm and the calibration coefficient, the simulated results would have greater discrepancies. The land evaporation with relatively high precision has been achieved when various methods are combined. Although this may not necessarily lead to better prediction ability of land evaporation and increased understanding of physical processes, uncertainties can be reduced by the integration of multiple remote sensing products (Jung et al., 2010; Mueller et al., 2013) and more complex data merging methods (Yao et al., 2014). The effectiveness of hydrometeorological monitoring can be improved by accurate quantification and further reduction of the uncertainty of water cycle variables especially land ET. Therefore, a variety of merging techniques have been introduced and applied to water cycle variables such as soil moisture and precipitation in recent years. Least-squares (Yilmaz et al., 2012) and maximization R (Kim et al.,

2015, 2018) techniques were proposed for satellite soil moisture products merging. In regard to precipitation data merging, geographically weighted regression algorithm (Xu et al., 2015), conditional merging (Baik et al., 2016), geographical difference analysis (Cheema and Bastiaanssen, 2012), geographic ratio analysis (Duan and Bastiaanssen, 2013) and Multi-Source Weighted-Ensemble Precipitation (MSWEP) method (Beck et al., 2017) have been widely used. As for land ET, several relevant studies have evolved through simple average to complex methods including the weighted average (Hobeichi

et al., 2018), reproducing flux observations combined with the original land evaporation product (Yao et al., 2017a), or seeking consistency between land evaporation and water cycle related products such as precipitation, runoff and land water storage (Aires et al., 2014; Munier et al., 2014). Various data merging methods, such as Kalman filtering algorithm (Pipunic et al., 2008; Liu et al., 2013), Bayesian Model Average (BMA) and Empirical Orthogonal Function (EOF), can improve regional ET estimation by merging multiple ET products (Yao et al., 2014, 2016; Feng et al., 2016; Zhu et al., 2016). Since

land ET is a complex variable coupling energy, hydrology and carbon budget, it is difficult to accurately determine the optimal conditional density function in BMA that determines the performance of the method (Yao et al., 2014). Simultaneously, their complexity affects the efficiency of calculating the weight of individual data set, which limits their wide application. Simple average (SA) (Ershadi et al., 2014) and simple Taylor skill's score (STS) merging (Yao et al., 2017b) have been adopted for global ET merging. However, SA assumes the same uncertainty in each data set, which is

actually unreasonable, and STS is highly dependent on the accuracy of individual data sets, which makes it highly quality-demanding for the data sets involved in the merging process. Khan et al. (2018) analyzed the sources of uncertainties for three different ET products using triple collocation (TC) method, which estimates the random error standard deviations for three datasets of the same variable according to statistical relations, and provides an uncorrelated absolute and relative error structure among data sets. Lastly, Jimenez et al. (2018) proposed an error variance unweighted merging method with the

local weights deduced according to the variance of the differences between the outliers of the flux tower and simulated land evaporation. The above merging methods effectively reduce the uncertainties of simulations by estimating the weight of multiple products to generate reliable merged products (Zhu et al., 2016). However, previous studies have mostly focused on the evaluation of ET simulation at regional scales and landscape (Yang et al., 2016).

Compared with the simple average method, Reliability Ensemble Average method (REA) extracts the most reliable

information from each model by minimizing the impact of "outliers" or underperforming models, subsequently reducing the uncertainty range in simulated changes, which also stands out in terms of computational efficiency (Giorgi & Mearns, 2002). These standards including the bias of the simulated ET from the reference and the distance of the simulated ET from the ensemble average are regional rather than global, as most models tend to show anomalous behavior or poor performance from one region to another. The REA method also produces a quantitative measure of reliability, indicating that the

simulations need to meet both criteria in order to improve the overall reliability of simulation changes. On the one hand, REA method considers model performance, that is, the ability of models reproducing current climate, which is defined by the difference between the simulation and observation. On the other hand, the model convergence, a factor to measure the

reliability of the models is taken into consideration as well. It is the distance between the changes of the given model and that of the ensemble average. The REA method has been widely used for meteorological variables such as precipitation and temperature. Giorgi & Mearns (2002) used the REA method to integrate the average seasonal temperature and precipitation of 22 land regions in the world under two emission scenarios simulated by 9 atmosphere-ocean circulation models in the late 21st century. However, there are few studies on the application of area-averaged grid-scale merging of long sequence model-based land evaporation data. This study aims to develop a long-term high-quality global land ET product using merging technology. Merging multiple single data sets is expected to reduce the uncertainties of land ET effectively. The merged product can provide a basis for water cycle research and global water resources management. Hence, the systematic and in-depth studies on land evaporation merging are urgently needed.

## 2 Data and methods

### 2.1 Data types

Three widely used land ET data sets were selected for merging, including the fifth-generation ECMWF Re-Analysis (ERA5; Hersbach et al., 2020), the second Modern-Era Retrospective analysis for Research and Applications (MERRA2; Gelaro et al., 2017), and Global Land Data Assimilation System Version 2 ET (GLDAS2; Sheffield & Wood, 2007). The differences in spatial and temporal resolution among the ET products were rescaled to a daily timescale and 0.25°, with the time span from 1980 to 2017. Global Land Evaporation Amsterdam Model (GLEAM; Miralles et al., 2011) was used as the reference data due to its relative independence from other data sets participating in the merging process. Ideally, in-situ data would be the first choice to be used as the reference data for the merging. However, these point-scale datasets are very scarce globally and only representative of their immediate locations. Therefore, area-averaged grid-scale estimates offer a better alternative at this scale. Additionally, GLEAM is not a complex terrestrial model as found in land models of ERA5, MERRA2 and GLDAS, but a set of algorithms dedicated to estimating terrestrial evaporation using retrieved satellite observations including soil moisture, vegetation optical depth and snow-water equivalent, a multi-source precipitation product and relies on only radiation and temperature inputs from reanalysis products (Martens et al., 2017). As such GLEAM offers a higher level of independence than the other products. Eddy Covariance (EC) ET was used to evaluate the merged product compared with other data sets involved in the merging process. Monthly GIMMS NDVI3g data with a spatial resolution of 0.25° from the Global Inventory Modeling and Mapping Studies (GIMMS) was used to study how the quality of land evaporation data sets change with vegetation in our study (Pinzon & Tucker 2014), with the time span from 1982 to 2014, which is available from http://ecocast.arc.nasa.gov/data/pub/gimms/3g/. The spatial and temporal resolutions of these ET datasets are shown in Table 1, which is briefly described below.

**Table 1. Summary of ET data sets involved in merging.**

| Name | ET schemes/ | Spatial | Temporal | Time span | Reference |
|------|-------------|---------|----------|-----------|-----------|

|  | land-surface schemes | resolution (degree) | resolution |  |  |
| --- | --- | --- | --- | --- | --- |
| GLEAM3.2a | Priestley-Taylor | 0.25×0.25 | daily | 1980-2017 | Miralles et al. (2011) Martens et al. (2017) |
| ERA5 | IFS | 0.25×0.25 | 1-hour | 1980-2017 | Hersbach et al. (2020) |
| MERRA2 | GEOS-5 | 0.625×0.5 | daily | 1980-2017 | Gelaro et al. (2017) |
| GLDAS2.0 & 2.1 | Noah | 0.25×0.25 | 3-hour | 1980-1999& 2000-2017 | Sheffield & Wood (2007) |

**2.1.1 Global Land Evaporation Amsterdam Model (GLEAM) ET**

GLEAM algorithm estimates land evaporation mainly based on the parameterized physical process. Stress conditions are parameterized as a function of dynamic vegetation information and available water in the root zone. In addition, the detailed parameterization of forest interception is one of its key features. Canopy interception loss, a component of land evaporation, is calculated by the daily precipitation using the parameters describing canopy storage, canopy coverage, and average precipitation and evaporation rate under saturated canopy conditions. It uses extensive independent remote sensing observations such as snow-water equivalent, vegetation optical depth and soil moisture as the basis for calculating land evaporation and its different components, including transpiration, bare-soil evaporation, interception loss, open-water evaporation and sublimation separately (Priestley and Taylor, 1972). The empirical parameters contained in this algorithm such as the evaporation stress factor, the latent heat of evaporation and the slope of the saturated water vapour-temperature curve have been obtained from the findings in different fields. On a global scale, GLEAM has been validated with the observations obtained from the eddy covariance instrument, indicating that it can be used to describe terrestrial ET in different ecosystems (Miralles et al., 2011). In addition, GLEAM is a long sequence data set predominantly based on remote sensing observations, and on occasion, reanalysis data. GLEAM is unlike traditional land models, such as found in ERA5, MERRA2 and GLDAS, in that it is driven by satellite observations to obtain evaporation estimates. The version of GLEAM here relies very little on reanalysis datasets (only radiation and temperature of ERA-Interim). Therefore, GLEAM has the most independence relative to the model-based products, which is selected as the reference data due to its relative independence. The version of the data set used in this study is 3.2a, which spans a 38-year period through 1980 to 2017, gridded with 0.25 degree. It is available from https://www.gleam.eu/.

### 2.1.2 The fifth-generation ECMWF Re-Analysis (ERA5) ET

Following ERA-15, ERA-40 and ERA-Interim, the fifth generation of ECMWF reanalysis data ERA5 has been released, which is envisioned to replace ERA-Interim reanalysis (Hersbach et al., 2020). Compared with ERA-Interim, some of the key climatic information of the EAR5 has been improved. The most updated version of the Earth System Model and data assimilation techniques used at ECMWF have been applied in ERA5, including more sophisticated parametrization of geophysical processes in comparison to the previous versions used in ERA-Interim. ERA5 covers from 1979 to the near real time period (on a regular basis), moreover, temporal and spatial resolution have been improved in ERA5, from 6-hourly in ERA-Interim to hourly, from 79 km to 31 km in the horizontal dimension and 60 to 137 in vertical levels. ERA5 has a better balance of global precipitation and evaporation (Albergel et al., 2018). Martens et al. (2020) evaluated surface energy partitioning in ERA5 especially including the latent heat fluxes using different reference datasets and modeling tools, with the analysis showing that there is lower absolute biases in the surface latent heat flux of ERA5 than ERA-Interim, though ERA5 still appears to overestimate the latent heat flux in most catchments. It is available from https://www.ecmwf.int/en/forecasts/datasets/reanalysis-datasets/era5/.

### 2.1.3 The second Modern-Era Retrospective analysis for Research and Applications (MERRA2) ET

MERRA2 (Gelaro et al., 2017) is an advanced atmospheric reanalysis data set, which absorbs mass of satellite data, including the new observation types such as hyperspectral radiation, microwave and aerosols. The MERRA2 is unique in modern reanalysis data set that contains aerosol data assimilation (Randles et al. 2017). Like ERA5, it combines satellite and more traditional weather observations with simulated atmospheric behavior, making an attempt to get the optimal possible estimation of the earth system state. MERRA2 is the second version of MERRA, which has undergone several major upgrades, including an observation-based precipitation bias correction (Reichle et al., 2017). The land surface model used in MERRA2 is the Catchment Land Surface Model (Koster et al., 2000), where land evaporation is calculated as part of the energy balance at the land surface. Hourly data with a 0.625× 0.5° spatial resolution are provided by the Goddard Earth Sciences Data and Information Service Center (DISC). Daily data from 1980 to 2017 has been used in this study. The accuracy of MERRA2 has been widely evaluated (Bosilovich et al., 2015; Gelaro et al., 2017), including water cycle variability and the global water balance (Bosilovich et al., 2017). It is available from https://goldsmr4.gesdisc.eosdis.nasa.gov/data/MERRA2/M2T1NXLND.5.12.4/.

### 2.1.4 Global Land Data Assimilation System ET (GLDAS) ET

GLDAS is a global high-resolution land modeling system based on North American Land Data Assimilation System (NLDAS). In order to generate better land surface products in different LSM, GLDAS generates the optimal field of surface state and flux by absorbing satellite and surface observation data, taking advantage of advanced land surface modeling (Rodell et al., 2004). Different LSMs are used including Mosaic, Noah, CLM, and VIC, where only Noah has continued until

now. Recently, there is more and more evidence to show that GLDAS-1.0 has serious discontinuities due to forcing data such as large precipitation and temperature errors in 1996 and 2000-2005 (Wang et al., 2016). Therefore, both daily and monthly land evaporation data of GLDAS2 combined with Noah LSM (GLDAS2-Noah) has been used in this study, whose spatial resolution is 0.25°×0.25°. GLDAS2 includes two data sets, specifically GLDAS-2.0 and GLDAS-2.1 where the simulations start from 1948 in GLDAS-2.0 and 2000 in GLDAS-2.1. The product is simulated with the Princeton University meteorological forcing data set (PUMFD), which has been corrected with observation-based products during the period of 1948-2010 (Sheffifield et al., 2006). Time period of GLDAS-2.0 from 1980 to 1999 and GLDAS-2.1 from 2000 to 2017 are selected in this study. Details of forcing data and description of the model are available on http://disc.Sci.GSFC.NASA.Gov/Hydrology. They are available from https://hydro1.gesdisc.eosdis.nasa.gov/data/GLDAS/GLDAS_NOAH025_3H.2.0/ and https://hydro1.gesdisc.eosdis.nasa.gov/data/GLDAS/GLDAS_NOAH025_3H.2.1/.

### 2.1.5 Eddy Covariance (EC) ET

Latent heat flux (LE) from 181 effective flux towers across the world is used to evaluate the performance of multiple data sets with different estimates, which is available from http://fluxnet.fluxdata.org/. Eddy covariance is a widely used energy flux measurement method that provides continuous measurement of interchange of water and energy (Mu et al., 2011). Measurements are masked with the provided quality flags in the data set archives. Since there is minimal impact of Ground-measured ET on days with strong precipitation on the verification results (Fig. S1), data on these days are not excluded in order to retain more site data. Figure 1 shows that the data availability of in situ data occurs over different periods. The data cover the period of 1992-2014, including at least 1 year reliable data. As shown in Fig. 1b, the periods vary from 1 to 19 years, with 14, 32, 32, 13 and 9 sites reporting 1, 2, 3, 4 and 5 years of data respectively, accounting for 55 percent of the sites. Ideally, a fair evaluation of the products could be done if the availability of the datasets fully overlapped. However, here, the limited available overlapping times of the insitu datasets makes their use quite inconsistence. Nonetheless, they are still useful since they offer an objective evaluation source. As such no filtering was applied to select specific data apart from the quality control applied. The observed land evaporation is calculated by latent heat flux in the following Eq. (1),

$$ET = \frac{LE}{\lambda},\tag{1}$$

where $\lambda$ is the conversion factor with the fixed value of 2.45MJ kg$^{-1}$. The flux towers are located in 11 land cover types including 17 Croplands (CRO), 23 Deciduous Broadleaf Forest (DBF), 1 Deciduous Needleleaf Forest (DNF), 13 Evergreen Broadleaf Forest (EBF), 42 Evergreen Needleleaf Forest (ENF), 34 Grasslands (GRA), 9 Mixed Forest (MF), 9 Open Shrublands (OSH), 8 Savannas (SAV), 19 Permanent Wetlands (WET) and 6 Woody Savannas (WSA) sites. In the rest of the paper, the different land cover types listed, which are representative of different ecosystem types, will be simply referred to as ecosystems.

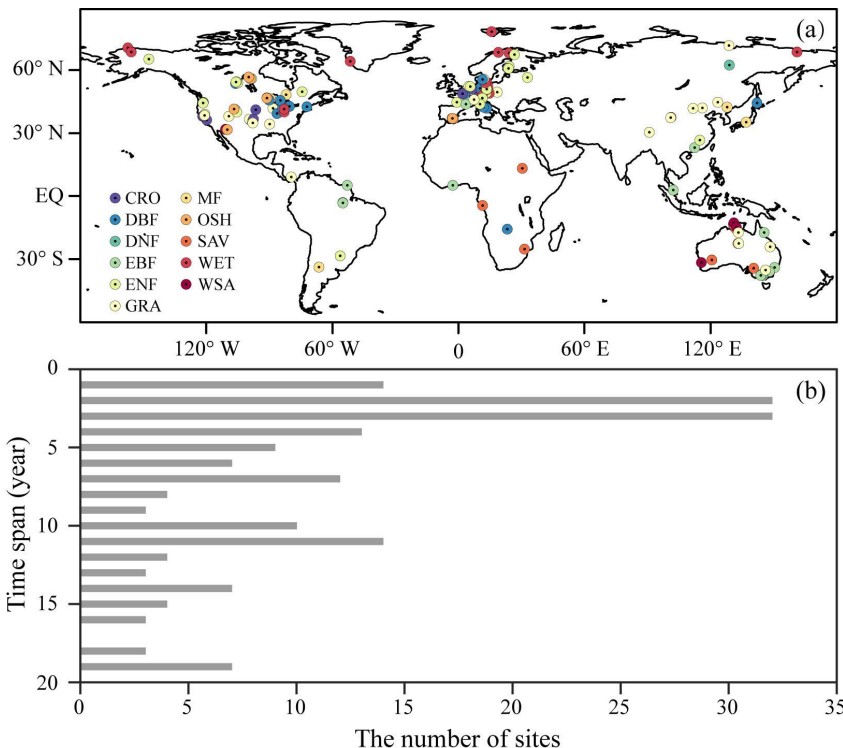

**Figure 1. a) Spatial distribution of 181 in-situ flux EC sites across the world and b) The number of sites for different time span.**

## 2.2 Methods

In this study, we investigated whether more accurate ET results could be obtained through the weighted combination of ERA5, GLDAS and MERRA2 estimation. Ideally, the weight assigned to each product should be based on an accurate description of the uncertainty during the merging process. Therefore, the three data sets have been weighted with respect to GLEAM, and the performance of the merged ET products has been studied at the selected sites. A set of weights need to be defined for weighted combination of three products, usually based on the individual uncertainty of each product. The simplest strategy used in previous studies is to assume that all three products have the same uncertainty, thus the merged product is a simple average of each product. A more detailed strategy used in this study is to weigh the product based on their uncertainties. The expected goal is to develop a product that minimizes Root Mean Square Deviation (RMSD), which we call optimal in the context of our merging strategy.

### 2.2.1 Coefficient of Variation

The coefficient of variation (CV), also known as the relative standard deviation in probability theory and statistics, is used to evaluate the consistency of multiple sets of reanalysis ET data. It is a statistic to measure the degree of variation in the data. The consistency decreases with the increase of CV. It is calculated from the following equation:

$$CV = \frac{S}{\bar{x}} \times 100\%, \tag{2}$$

where S represents the standard deviation and $\bar{x}$ represents the average of multiple sets of reanalysis ET data in each pixel.

This approach is superior to standard deviation in terms of evaluating consistency, which can eliminate the influence of different units and/or average on the degree of variation of two or more data. In this paper, multiple sets of reanalysis land evaporation data are blended into a single product based on their performance and convergence criteria.

### 2.2.2 Reliability Ensemble Averaging

Reliability Ensemble Averaging (REA) method (Giorgi and Mearns, 2002; Xu et al., 2010) was used to combine multiple
sets of model-based ET data into a single product. Two reliability criteria were considered in the method: model performance and model convergence, in other words, the model's performance in reproducing the current climate and the convergence of simulated values between models.

In our REA method, the average ET is given by a weighted average of all the ensemble members.

$$\widetilde{ET} = \tilde{A}(ET) = \frac{\sum_i R_i ET_i}{\sum_i R_i}, \tag{3}$$

where $\tilde{A}$ represents the REA averaging and $R_i$ represents the model reliability factor defined as:

$$R_i = \left[ (R_{B,i})^m \times (R_{D,i})^n \right]^{[1/(m \times n)]}$$
$$= \left\{ \left[ \frac{\varepsilon_{ET}}{abs(B_{ET,i})} \right]^m \left[ \frac{\varepsilon_{ET}}{abs(D_{ET,i})} \right]^n \right\}^{[1/(m \times n)]}, \tag{4}$$

The merging is extended to pixels rather than just flux tower level, based on the selection of independent GLEAM as the reference to compensate for the limited EC measured ET. $R_{B,i}$ and $R_{D,i}$ in Eq. (4) are measures of the model performance and convergence criteria respectively. $R_{B,i}$ is a factor to measure the reliability of the model through the bias ($B_{ET,i}$) between the simulated ET and the reference, that is, the larger the bias the lower the reliability of the model. $R_{D,i}$ is a factor to measure the reliability in the aspect of the distance ($D_{ET,i}$) between the simulated ET and the ensemble average, that is, the higher the
distance the lower the reliability of the model.

The parameters m and n in Eq. (4) are used to measure the relative importance of the two criteria. In this work, assuming the importance of the two criteria is equal, m and n were assigned with 1. However, if the two criteria are given different weights, they may be different. The parameter $\epsilon$ in Eq. (4) is a measure of natural variability in 38-yr ET. In order to calculate $\epsilon$, we estimated moving averages after linearly detrending the 38-yr time series data in each pixel. Then, the
differences between maximum and minimum of the moving averages are computed as $\epsilon$. In addition, when B and D is less

than ϵ, $R_B$ and $R_D$ are set to 1 respectively. In essence, Eq. (4) indicates that the model is reliable when the bias and the distance from the ensemble average are within the limits of natural variability, where RB=RD=R=1. With the bias or distance growing, the reliability of a given model decreases.

However, the performance of a data set varies across all time points in a certain region, with some time performing better and some worse. This will be taken into account in the next release of the data.

The specific merging steps are as follows:

Step 1: Select the best climatology according to the root mean square deviation (RMSD) between each data set and EC measured ET.

Step 2: Calculate the anomalies of each data set participating in the merging and the reference data.

Step 3: Merge the anomalies.

Step 4: Add the best climatology to the merged anomalies to get the final merged product.

### 2.2.3 Validation Criteria

The error metrics of Pearson correlation coefficient (R), root mean square deviation (RMSD) and unbiased root-mean-square deviation (ubRMSD) were used to verify the blending product. The statistic values are defined as follows:

$$R = \frac{\sum_{i=1}^{n}(M_i - \bar{M})(ref_i - \overline{ref})}{\sqrt{\sum_{i=1}^{n}(M_i - \bar{M})^2}\sqrt{\sum_{i=1}^{n}(ref_i - \overline{ref})^2}} \, , \tag{5}$$

$$RMSD = \sqrt{n^{-1}\sum_{i=1}^{n}(M_i - ref_i)^2} \, , \tag{6}$$

$$BIAS = n^{-1}\sum_{i=1}^{n}(M_i - ref_i) \, , \tag{7}$$

$$ubRMSD = \sqrt{RMSD^2 - BIAS^2} \, , \tag{8}$$

where $n$ represents the sample size; $M_i$ and $ref_i$ respectively represents multiple sets of reanalysis ET data and reference data at time $i$. $\bar{M}$ and $\overline{ref}$ represent the average of $M_i$ and $ref_i$.

### 3 Results and Discussion

The consistency of the three data sets has been illustrated in Fig. 2, where Fig. 2a-c shows the differences of land evaporation between each data set and the mean of the three products. In the high latitudes of the northern hemisphere, GLDAS-Noah2 ET is more than 20% higher than the mean of the three products, and ERA5 ET is almost the same as the

mean, while MERRA2 ET is more than 20% lower than the mean. In the middle latitudes, ERA5 ET is more than 20% higher than ensemble mean, while GLDAS-Noah2 ET is more than 20% lower than it. As for MERRA2 ET, there are a few areas higher than the mean of the three products. In the western part of South America and parts of Oceania in the southern hemisphere, ERA5 ET is higher than the mean of the three products while GLDAS-Noah2 is lower than it, MERRA2

showing no significant difference. In these regions, the three data sets are significantly different, indicating that the estimation of land evaporation is of great uncertainty and low consistency. In order to reduce the risk of inaccuracy in land evaporation merging, CV is used to select regions with high consistency. The CV analysis aims to evaluate the relative systematic differences between the three model products. Since it does not take the reference data into account, it does not directly translate into the merging scheme. Nonetheless, it serves as an added check to obtain optimum consistencies in the

merging process for higher skill in the merged data. Not surprisingly, in the north of North America, west of South America, desert regions of mid-latitude Africa and Asia, CV is above 0.8, indicating relatively low consistency and high risk, thus these regions are excluded from the merging region. In essence, these excluded areas are concentrated in hyper-arid areas where some methods for estimating land evaporation are not applicable (Goya and Harmsen, 2014). If the data we used for merging are highly different from each other and none of them are close to the reference data, merging in these regions does

not make sense. For the overall reliability of the merged product, we excluded these areas that might be highly uncertain.

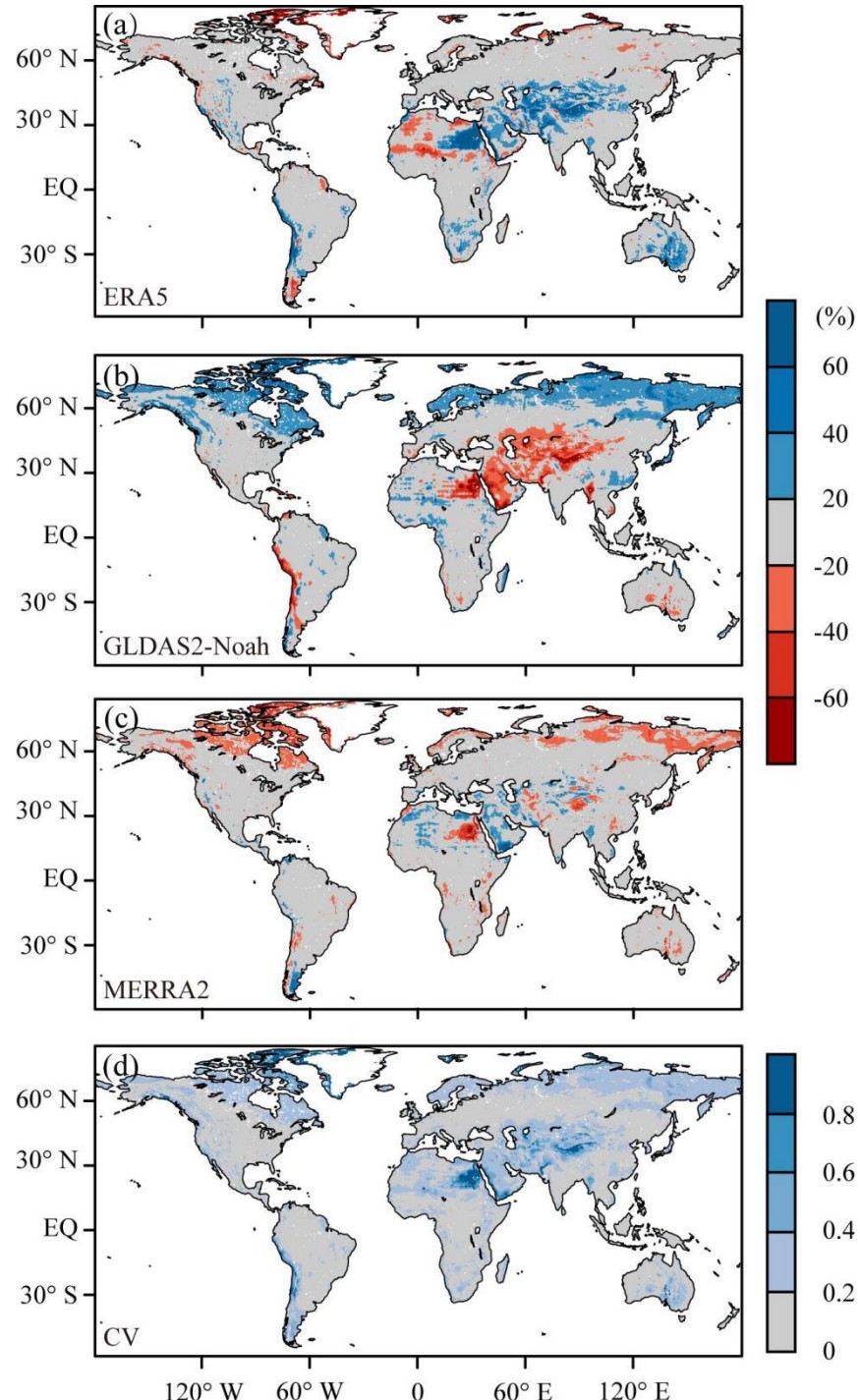

**Figure 2. a-c) The percentage of the difference between ERA5, GLDAS2-Noah, MERRA2 and average of the three data sets, d) Variable Coefficient of the three data sets.**

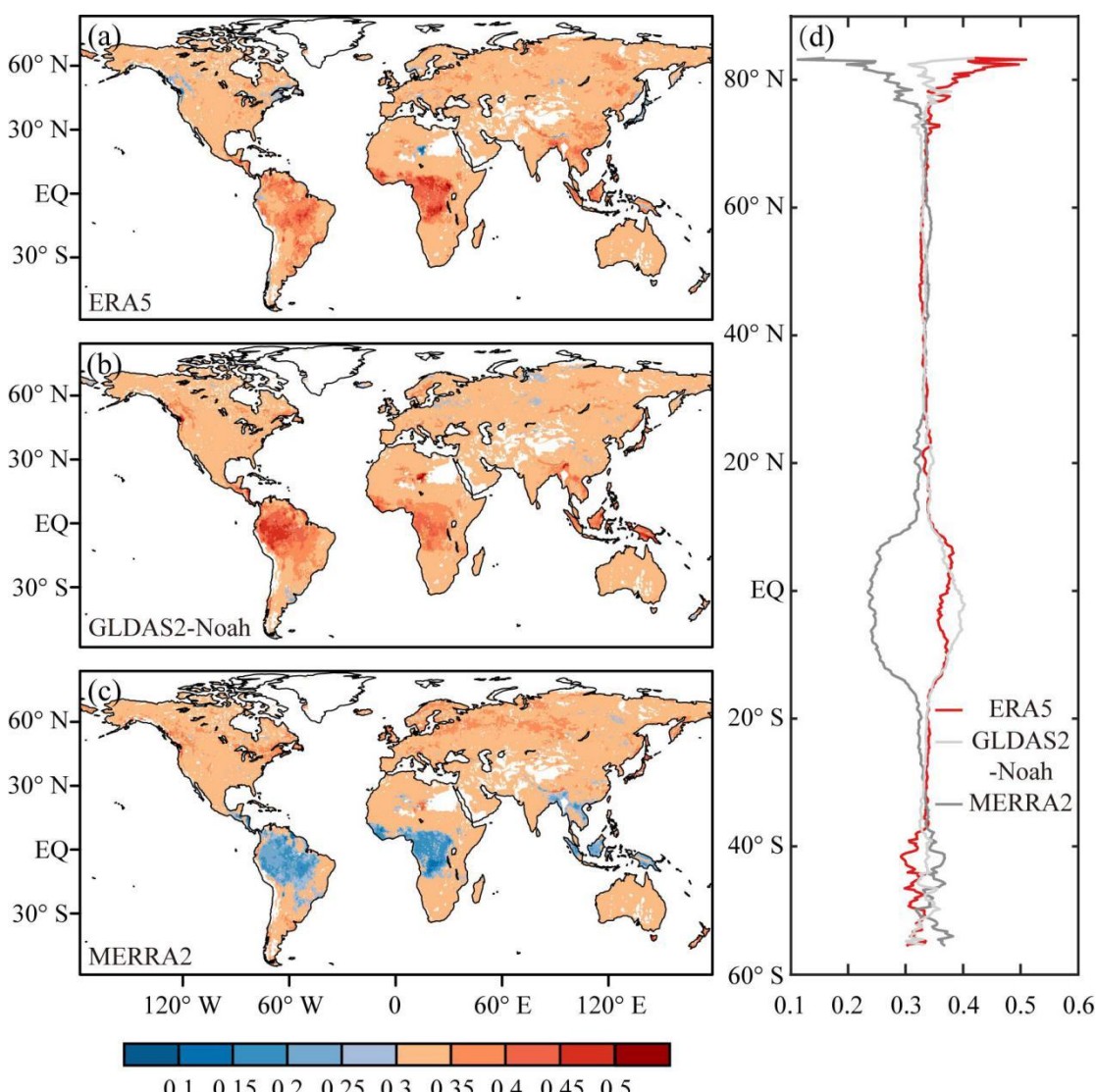

**Figure 3. a-c) Spatial distribution of weights, and d) latitudinal distribution of weights.**

The spatial distribution of weights has been depicted in Fig. 3, representing the contribution of each data set to the merged product. It is not difficult to find that the weights are within the scope of 0.3 ~ 0.35 in most regions, indicating that the contributions of the three data sets in these regions are basically the same. In the Amazon Plain near the equator, the Congo Basin and the border between Oceania and Asia, the weights of multiple data sets vary greatly. The weights of MERRA2 ET in these regions are below 0.3, while ERA5 ET and GLDAS-Noah2 ET are above 0.35, indicating that MERRA2 ET contributed less than the other two data sets in these regions. GLDAS-Noah2 ET is found to contribute greatly to the Amazon Plain and the border between Oceania and Asia, while in the Congo Basin ERA5 contributes the most. Zonal banded weights of the three data sets are presented in Fig. 3a-c. Three curves are shown in Fig. 3d, which describe the

contributions of each data set at different latitudes. Obviously, the contribution of MERRA2 ET is less than that of the other two data sets near the equator, further GLDAS-Noah2 ET contributes slightly more than ERA5 ET. In the high latitudes of the Northern Hemisphere, the difference in contributions among the three data sets is greatest, where the contribution of ERA5 ET is the highest. In the Southern Hemisphere, south of 40S, MERRA2 makes a higher contribution than the other two data sets, with little difference in the contribution of the three data sets in other regions.

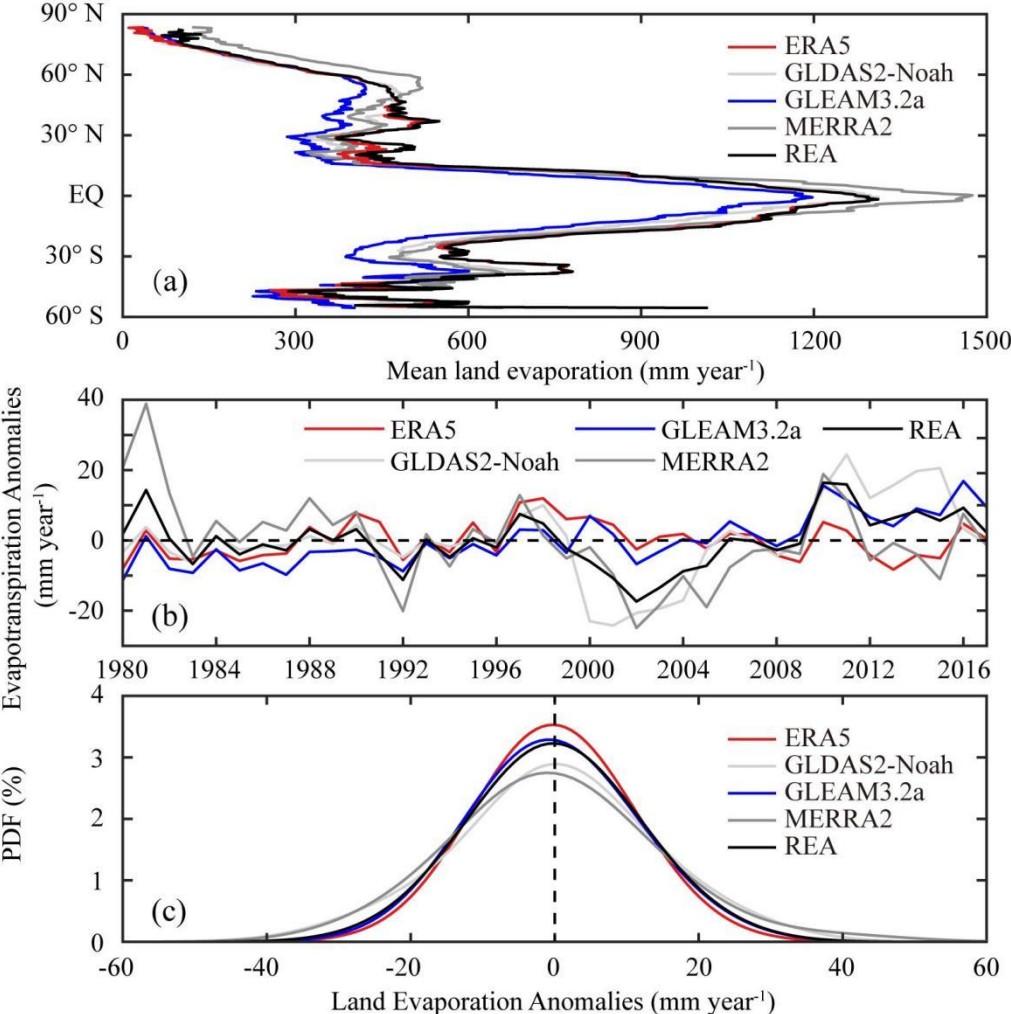

Figure 4. a) Latitudinal distribution of mean land evaporation from five data sets, b) time series (1980-2017), and c) probability distribution of annual land evaporation anomalies from five ET products. The bandwidth of the kernel smoothing window was set to 10.

The latitudinal distribution of the multiyear mean land evaporation of five products is shown in Fig. 4a. General consistency in spatial pattern is shown in spite of differences in intensity. However, large differences appear in the interannual variation of these products (Fig. 4b), though the long-term trends generally show good consistency. Anomalies for all data sets are shown in Fig. 4b. The comparison reveals similar temporal variations of these data sets over most periods. The land

evaporation time series of all data sets reach their low ebb between 2000 and 2005. Most of them peak between 2010 and 2011, except for MERRA2 ET arising in 1981 and ERA5 ET in 1998. It is conspicuous that the fluctuation of the merged product and four individual data sets are relatively small in the first half period of the 38 years, while in the second half one, relatively large fluctuation could been observed from all data sets except ERA5 ET. The merged product as well as four individual data sets show significant decreasing trends from 1997 to 2002 and increasing trends from 2002 to 2010. Fig. 4b also shows that the differences between REA and the other products varies with time in the interannual variation. That is, the errors are not stationary. Nonetheless, their long term trends generally show good consistency. The obvious differences between the probability density distributions of multiple data sets are clearly visible. In general, the consistency between the merged product and GLEAM ET is relatively better, which may be greatly related to GLEAM as the reference data in the merging process. Due to the discrepancies in the driving data and calculation formulations for land evaporation, anomalies vary from data to data.

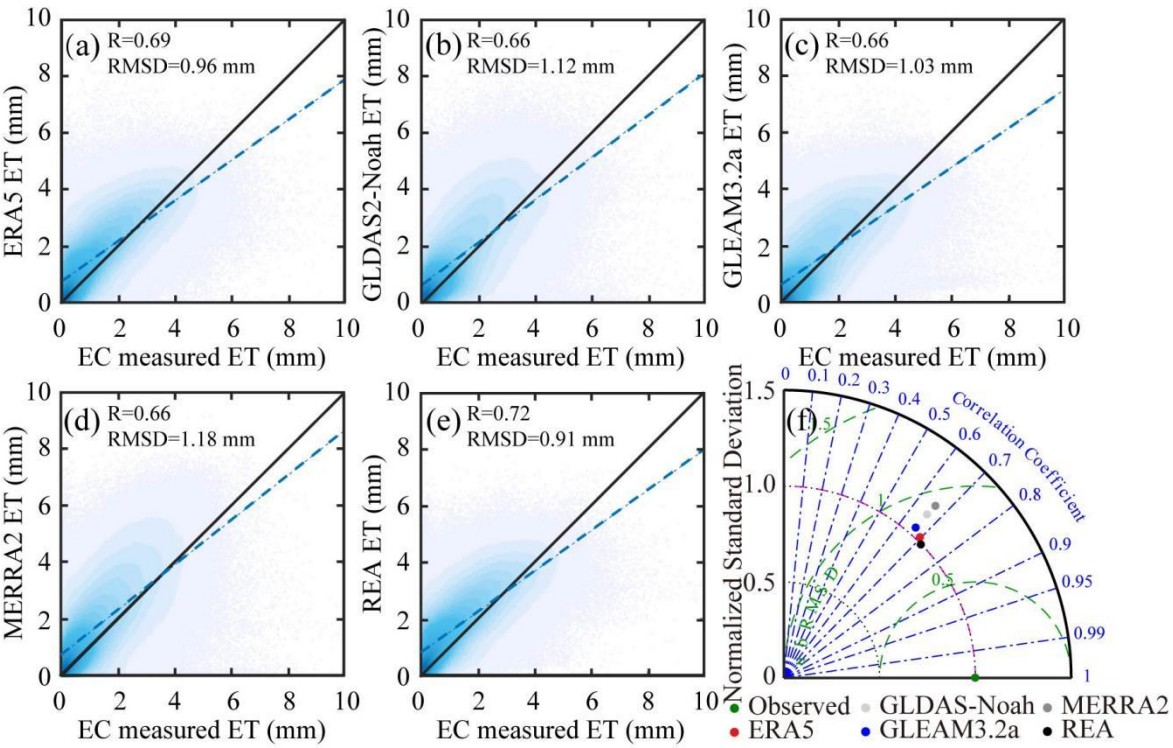

**Figure 5. a-e) Scatter plots and f) Taylor diagrams of daily Ground-measured ET and ET from different products. Linear fits are plotted in blue and the 1:1 line is depicted. The correlation coefficients (R) have passed the 5% significance test.**

Figure 5a-e shows the scatter diagram of multiple data sets and station-observed data at daily scale. Relatively more points are found concentrated above the 1:1 line, which makes it clear that land evaporation is somewhat overestimated. Among the five data sets, the correlation coefficient between REA and station-observed data is the highest, reaching 0.72, followed by ERA5, other three data sets are basically the same. Therefore, the correlation coefficient is significantly improved through

REA data merging, indicating more consistent changes between REA and station-observed data. Similarly, as for RMSD, REA is the smallest, only 0.91 mm. Therefore, the deviation between REA data merging product and station-observed data is significantly reduced compared with other data, indicating that the accuracy is greatly improved. The Taylor chart in Fig. 5f describes the ratio of standard deviation, correlation coefficient and unbiased root-mean-square deviation between five data sets and station-observed data under daily scale, with the ratio of standard deviation representing the ratio between each data

set and station-observed data. The ratio of standard deviation of REA is the smallest, nearly 1, indicating almost the same with station-observed data and the smallest fluctuation among all data sets considered. The ubRMSD values of multiple data sets are slightly smaller than RMSD respectively, while the rank of them remains the same, specifically REA < ERA5 < GLEAM3.2a < GLDAS-Noah2 < MERRA2. The correlation coefficient between REA and station-observed data is the highest, meanwhile the ratio of standard deviation, RMSD and ubRMSD are the lowest, indicating that REA performs

optimally under all assessment criteria.

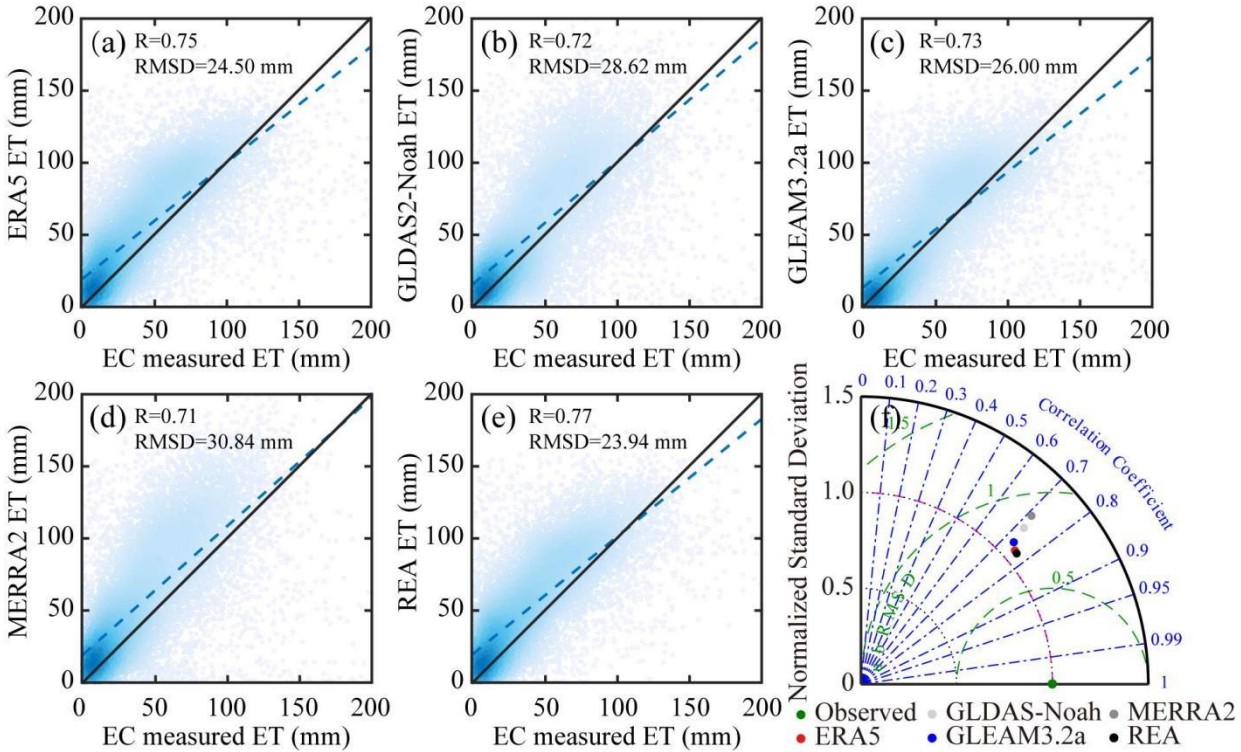

**Figure 6. a-e) Scatter plots and f) Taylor diagrams of monthly Ground-measured ET and ET from different products. Linear fits are plotted in blue and the 1:1 line is depicted. The correlation coefficients (R) have passed the 5% significance test.**

Figure 6 verifies the quality of multiple land evaporation data sets at monthly scale based on station-observed data. The

340 correlation coefficient between these data sets and station-observed data under monthly scale is higher than that under daily scale, with REA the highest, reaching 0.77. Among the five data sets, RMSD between REA and station-observed data is the smallest, only 23.94 mm. It can be seen from the Taylor chart that the ratio of standard deviation of each data set under

monthly scale is larger than that under daily scale, indicating the increase of fluctuation. Like the daily scale, REA performs optimally under all assessment criteria.

**Table 2. The verification results including R and RMSD between daily Ground-measured ET and daily ET from different products in different ecosystems. Values in bold indicates the highest quality.**

| Ecosystem type | ERA5 | | MERRA2 | | GLDAS | | GLEAM | | REA | |
|---|---|---|---|---|---|---|---|---|---|---|
| | R | RMSD | R | RMSD | R | RMSD | R | RMSD | R | RMSD |
| CRO | **0.66** | 1.24 | 0.55 | 1.42 | 0.60 | 1.48 | 0.60 | **1.22** | 0.60 | 1.38 |
| DBF | 0.76 | **1.06** | 0.71 | 1.23 | 0.74 | 1.19 | 0.67 | 1.16 | **0.77** | 1.07 |
| DNF | **0.81** | **0.55** | 0.73 | 0.75 | 0.77 | 0.73 | 0.64 | 0.75 | 0.80 | 0.62 |
| EBF | **0.72** | **1.08** | 0.61 | 1.59 | 0.58 | 1.36 | 0.70 | 1.11 | 0.65 | **1.08** |
| ENF | 0.66 | 1.03 | 0.66 | 1.21 | 0.67 | 1.05 | 0.66 | 1.04 | **0.73** | **0.88** |
| GRA | 0.72 | 1.05 | **0.77** | 1.11 | 0.70 | 1.09 | 0.73 | 0.96 | **0.77** | **0.94** |
| MF | 0.77 | **1.05** | **0.79** | 1.37 | 0.70 | 1.23 | 0.70 | 1.12 | 0.74 | 1.12 |
| OSH | 0.43 | 1.00 | 0.47 | 0.92 | 0.46 | 0.96 | 0.33 | 1.15 | **0.50** | **0.88** |
| SAV | 0.61 | 1.23 | 0.62 | 1.40 | 0.63 | 1.22 | 0.58 | 1.25 | **0.66** | **1.16** |
| WET | **0.57** | **1.40** | 0.44 | 1.66 | 0.47 | 1.56 | 0.52 | 1.44 | 0.46 | 1.59 |
| WSA | 0.68 | 1.24 | 0.63 | 1.46 | 0.64 | 1.17 | **0.72** | **1.11** | 0.70 | 1.13 |

Verification of ET products from different ecosystems has been conducted in order to further evaluate their performances. Table 2 describes quantitatively the performances of the ET products in 11 ecosystem types of site on a daily scale from two indicators, R and RMSD. The values in bold print indicate the best performance of the four products. The results demonstrate that no individual product performs best across all ecosystems. For 42 ENF, 34 GRA, 9 OSH, 8 SAV sites, REA has higher R and lower RMSD than individual products. For 23 DBF and 13 EBF sites, REA has a optimal R or RMSD; specifically with the highest R of 0.77 and second lowest RMSD of 1.07 mm per day for DBF, and the lowest RMSD of 1.08 mm per day and the second highest R of 0.65 for EBF. REA performs worse than at least one individual product at 63 other sites. Specifically, REA has a lower R of 0.60 than ERA5, and a higher RMSD of 1.38 mm per day than GLEAM and ERA5 at 17 CRO sites. For 1 DNF site, REA has lower R of 0.80 and higher RMSD of 0.62 mm per day than ERA5. For 9 MF sites, REA has a lower R of 0.74 than ERA5 and MERRA2, and a higher RMSD of 1.12 mm per day than ERA5. For 19 WET sites, REA has a lower R of 0.46 and a higher RMSD of 1.59 mm per day than ERA5 and GLDAS. For 6 WSA sites, REA has lower R of 0.70 and higher RMSD of 1.13 mm per day than GLEAM. Generally, ERA5, MERRA2, GLEAM and REA show the best performance respectively in four (including CRO, DNF, EBF and WET), two (GRA and MF), one (WSA) and five (DBF, ENF, GRA, OSH and SAV) ecosystems in terms of R. Based on RMSD, both ERA5 and REA performed best in five ecosystems (with the former including DBF, DNF, EBF, MF and WET, and the latter including EBF, ENF, GRA, OSH and SAV). REA does not perform best across all ecosystems, however, it avoids the worst performance in any

ecosystem. Taylor Diagram results of daily Ground-measured ET and ET from the different products in 11 ecosystems are put in support information (Fig. S2).

**Table 3. The verification results including R and RMSD between monthly Ground-measured ET and monthly ET from different products in different ecosystems. Values in bold indicates the highest quality.**

| Ecosystem type | ERA5 | | MERRA2 | | GLDAS | | GLEAM | | REA | |
|---|---|---|---|---|---|---|---|---|---|---|
| | R | RMSD | R | RMSD | R | RMSD | R | RMSD | R | RMSD |
| CRO | 0.71 | 31.84 | 0.58 | 38.33 | 0.64 | 39.59 | 0.66 | **30.85** | **0.73** | 31.14 |
| DBF | 0.84 | 26.61 | 0.77 | 32.40 | 0.83 | 30.27 | 0.75 | 28.43 | **0.86** | **26.17** |
| DNF | **0.93** | **10.62** | 0.84 | 18.63 | 0.91 | 18.96 | 0.85 | 14.92 | 0.91 | 11.01 |
| EBF | 0.81 | 25.85 | 0.71 | 40.32 | 0.69 | 32.21 | **0.84** | **22.79** | 0.78 | 27.06 |
| ENF | 0.74 | 25.01 | 0.72 | 31.69 | **0.76** | 25.38 | **0.76** | 24.24 | **0.76** | **23.91** |
| GRA | 0.77 | 27.11 | **0.83** | 28.59 | 0.77 | 26.54 | 0.81 | **22.06** | 0.77 | 27.53 |
| MF | 0.83 | 27.57 | **0.85** | 37.98 | 0.75 | 32.90 | 0.79 | 27.71 | 0.83 | **26.99** |
| OSH | 0.45 | 25.34 | 0.51 | **23.39** | 0.52 | 23.75 | 0.27 | 32.57 | **0.53** | 24.54 |
| SAV | 0.65 | 32.05 | 0.65 | 38.08 | 0.67 | 31.88 | 0.59 | 34.33 | **0.68** | **31.45** |
| WET | 0.61 | 38.30 | 0.46 | 46.92 | 0.49 | 43.78 | 0.55 | 40.38 | **0.64** | **37.35** |
| WSA | 0.73 | 33.71 | 0.68 | 38.54 | 0.72 | 28.62 | **0.77** | **28.12** | 0.73 | 34.49 |

Similar to Table 2, Table 3 shows the performance of ET products in different ecosystems on a monthly scale. Compared with daily scale, the performance of each product has changed, among which all of the R becomes higher. REA has higher R and lower RMSD than individual products at 23 DBF, 42 ENF, 8 SAV and 19 WET sites. It has an optimal R or RMSD at 370 17 CRO, 9 MF and 9 OSH sites. At other 64 sites, REA has a worse performance than at least one individual product. For 1 DNF site, REA has lower R and higher RMSD than ERA5. For 13 EBF sites, REA has lower R and higher RMSD than GLEAM and ERA5. For 34 GRA sites, ERA5 has a lower R of 0.77 than MERRA2 and GLEAM, and a higher RMSD of 27.53 mm per month than GLEAM, ERA5 and GLDAS. For 6 WSA sites, REA has a lower R of 0.73 than GLEAM, and a higher RMSD of 34.49 mm per day than GLEAM, GLDAS and ERA5. Similar to the daily scale, REA does not have a 375 better performance than any individual product in all ecosystems, but is superior to at least one individual one. Similarly, the Taylor charts of monthly Ground-measured ET and ET from the different products in 11 ecosystems are put in support information (Fig. S3).

**Table 4. The verification results including R and RMSD between daily Ground-measured ET and daily ET from different products in different seasons. Values in bold indicates the highest quality.**

| Season | ERA5 | | MERRA2 | | GLDAS | | GLEAM | | REA | |
|---|---|---|---|---|---|---|---|---|---|---|
| | R | RMSD | R | RMSD | R | RMSD | R | RMSD | R | RMSD |
| MAM | **0.63** | **1.12** | 0.62 | 1.44 | 0.58 | 1.21 | 0.57 | 1.19 | **0.63** | 1.15 |

| | | | | | | | | | |
|---|---|---|---|---|---|---|---|---|---|
| JJA | **0.44** | **1.45** | 0.41 | 1.77 | 0.42 | 1.69 | 0.40 | 1.47 | **0.44** | 1.47 |
| SON | 0.64 | 0.96 | 0.61 | 0.93 | 0.60 | 0.93 | 0.60 | 0.91 | **0.65** | **0.84** |
| DJF | 0.78 | 0.77 | 0.75 | 0.83 | 0.74 | 0.77 | 0.76 | 0.77 | **0.79** | **0.70** |

In addition to different ecosystems, seasonal validation has been carried out to try to find out how each ET product performs in different seasons. Table 4 shows the performance of five ET products in different seasons on a daily scale. In general, REA has a better performance than individual products in autumn and winter, while in spring and summer it has a worse performance than ERA5. The R of REA for all seasons varies from 0.44 to 0.79, which remains optimal. In spring and summer, its R is as high as ERA5, and RMSD is second only to the best-performing ERA5. In addition, MERRA2, GLDAS

and GLEAM perform similarly. In terms of the whole year, the R of all products is higher in winter and lower in summer than other seasons. Similarly, RMSD is the highest in summer and the lowest in winter, which is mainly caused by the large variation and absolute value of land ET in summer. The Taylor charts of daily Ground-measure ET and ET from the different products in four seasons are put in support information (Fig. S4).

**Table 5. The verification results including R and RMSD between monthly Ground-measured ET and monthly ET from different**
**products in different seasons. Values in bold indicates the highest quality.**

| Season | ERA5 | | MERRA2 | | GLDAS | | GLEAM | | REA | |
|---|---|---|---|---|---|---|---|---|---|---|
| | R | RMSD | R | RMSD | R | RMSD | R | RMSD | R | RMSD |
| MAM | 0.69 | 28.93 | 0.68 | 39.04 | 0.65 | 30.54 | 0.63 | 30.69 | **0.71** | **28.51** |
| JJA | 0.42 | 37.67 | 0.38 | 48.15 | 0.41 | 44.89 | 0.40 | 37.26 | **0.45** | **36.71** |
| SON | 0.71 | 23.66 | 0.67 | 23.08 | 0.66 | 22.58 | **0.72** | **20.41** | 0.72 | 23.59 |
| DJF | **0.85** | 18.18 | 0.84 | 19.24 | 0.84 | 17.14 | **0.86** | **16.52** | 0.85 | 18.35 |

Compared with the daily scale, the performance of REA varies greatly in different seasons at the monthly scale (Table 5). In spring and summer, REA performs better than all individual products. While in autumn, REA has a higher R of 0.72 than individual products, and slightly lower RMSD of 23.59 mm per month than ERA5. In winter, REA has a higher R of 0.85 than MERRA2 and GLDAS, and a lower RMSD of 18.35 mm per day than MERRA2. Like the daily scale, the performance

of all products is still better in winter and worse in summer. Although the performances of REA at the daily and monthly scales vary in each season, the R is always higher than individual products except in winter, indicating the highly consistent of variation of REA with the observations. As well, the Taylor charts of monthly Ground-measure ET and ET from the different products in four seasons are put in support information (Fig. S5). Tables 4 and 5 demonstrate that the errors, in both the REA and the other products, vary across different time points, suggesting a non-stationtionarity of the uncertainties.

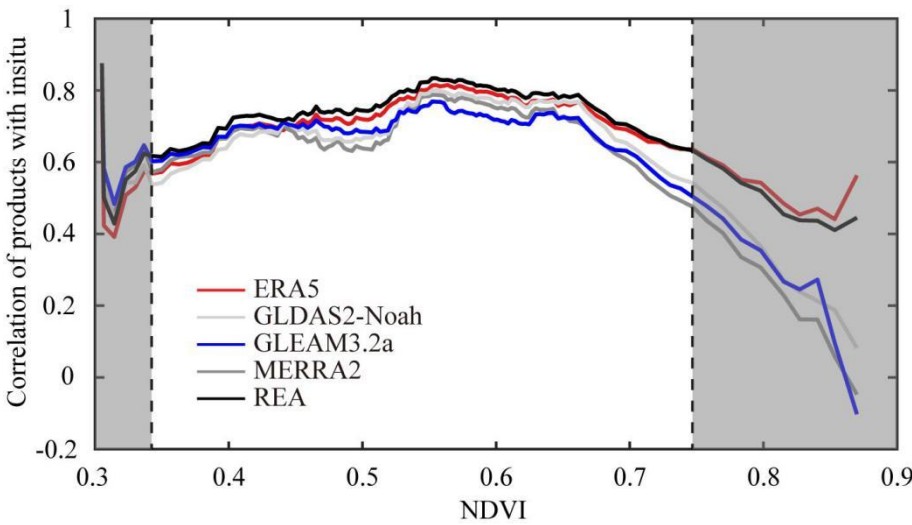

**Figure 7. Evaluation over different vegetation densities (NDVI) of the merged product based the correlation coefficient with station-observed data.**

Previous studies show that there is a close relationship between the quality of land evaporation and vegetation (Miralles et al., 2016). The correlation coefficients between multiple data sets and station-observed data under different vegetation

conditions (0.3 to 0.9) are compared in order to understand how the quality of these data sets change with vegetation. The results show that the quality of all data sets first increases and then decreases with the increase of vegetation density, with the highest quality captured when NDVI is around 0.55. It is worth noting that all data sets are in a good quality range with a correlation of more than 0.6 when NDVI is between 0.4 and 0.7. It is well known that vegetation index saturation poses potential issues. Generally speaking, NDVI is likely to become saturated over a dense canopy for forested areas, and

becomes saturated rapidly for vegetation with a nearly closed canopy (Liu et al., 2011). Based on the analysis of hyperspectral data, it is found that there is an obvious saturation problem in the relationship between LAI and NDVI, that is, when LAI exceeds 2, NDVI asymptotically reaches the saturation level (Haboudane et al., 2004). When biomass reaches a certain level, NDVI is not sensitive to changes in biomass (Huang et al., 2021). Dynamic vegetation is not used in these models, resulting in lower data quality with dense vegetation. Therefore, vegetation index saturation at high amounts of

NDVI results in a decrease in the quality of these datasets at high vegetation density. As shown in Fig. 7, the quality of each data set is relatively low and shows a rapid decline with the increase of vegetation density when NDVI is greater than 0.7, the case of optimal conditions for vegetation growth. In addition, a lot of remote sensing data have been used in GLEAM, such as satellite soil moisture, which is not of high quality when the vegetation density is high, affecting the quality of the final output. Further, errors in GLEAM will affect the merged product because GLEAM acts as the reference data. The

merged product demonstrates the ability to capture land evaporation dynamics in a wide range of vegetation densities, which performs best in all data sets when NDVI ranges from 0.34 to 0.75.

There are unique advantages and limitations of the existing land ET data sets for specific land cover types, however, quite few are globally suitable for meteorology and hydrology. Specific land cover classifications are assigned for each model, leading to the use of land cover classification from different sources bringing about discrepancies in the estimation of land ET. In different climatic regions, the performances of land ET products vary from the model responses. Feng et al. (2018) analyzed correlation between land ET estimated based on the Budyko hypothesis and reanalysis ET products, with results showing that great problems existed in MERRA2 when describing annual variation and long-term trend of land ET in China, mainly due to the higher variance amplitude of MERRA2 than that of other reanalysis products. Further, great uncertainty were captured in semiarid, semihumid and humid regions according to MERRA2. However, Dembele et al. (2020) found that MERRA2 was still one of the best data sets in estimating land ET in Volta River basin from 2003 to 2012 despite its low spatial resolution, which was probably due to the compensation of high temporal for low spatial resolution. In contrast, ERA5 ET behaved poorly in the region. Baik et al. (2018) studied the uncertainty of four widely used ET products (GLDAS2, GLEAM, MOD16 and MERRA) in the dry continent Australia during 2005-2014, finding a good consistency of GLDAS2 in arable land. GLEAM performed well in forest and savanna, while GLDAS2 showed the highest correlation in farmland, grassland, and shrub. However, GLDAS2 and GLEAM ET were found overestimated in most climatic regions and land cover classifications. The difference between model input variables can effectively explain the difference between estimated ET (Yao et al., 2017b). Specifically for Amazon tropical forests, MERRA2 tended to overestimate daily net radiation and incident solar radiation, while GLDAS2 tended to overestimate daily radiation and underestimate incident solar radiation (Gomis-Cebolla et al., 2019).

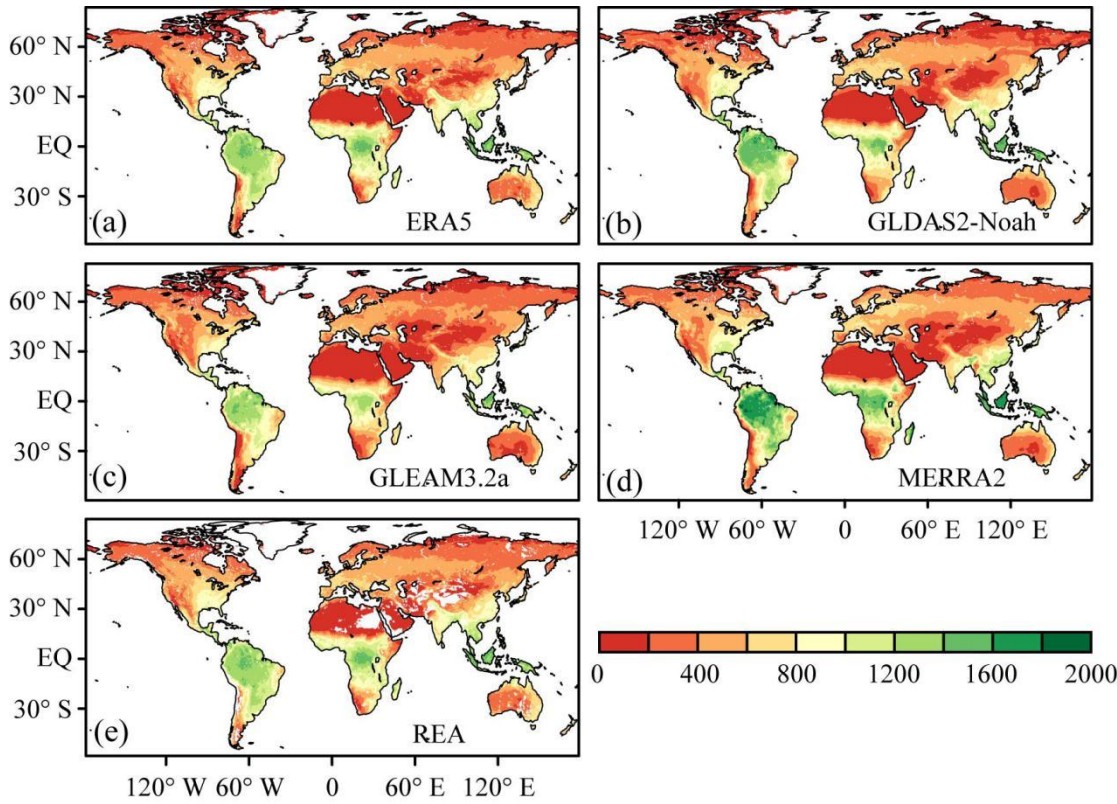

**Figure 8. Spatial distribution of annual mean land evaporation for the period 1980-2017 (unit: mm).**

Figure 8 depicts the spatial distribution of annual land evaporation, which seems to be relatively consistent of the five data sets. The regions with high land evaporation are found concentrated near the equator, generally very wet regions, including the Amazon Plain in the north of South America, the Congo Basin in central Africa and the border between Asia and Oceania, where the rainfall is usually over 1000 mm per year. Extremely low land evaporation is found concentrated in very dry desert and permafrost regions, including the Sahara and Arabian deserts in the north of Africa, the Taklimakan, Turkish, Iranian and Indian deserts in central Asia, and the permafrost regions in the north of North America and Eurasia, where the rainfall is under 200 mm per year. In comparison with REA, the measurements of MERRA2 and GLDAS-Noah2 are found significantly higher in the very wet regions near the equator, and basically the same in other regions, while GLEAM3.2a is slightly lower in the very wet equatorial regions and significantly lower in the very dry regions of central Asia and the west of North America. The spatial distributions of ERA5 and REA are found to be the most consistent.

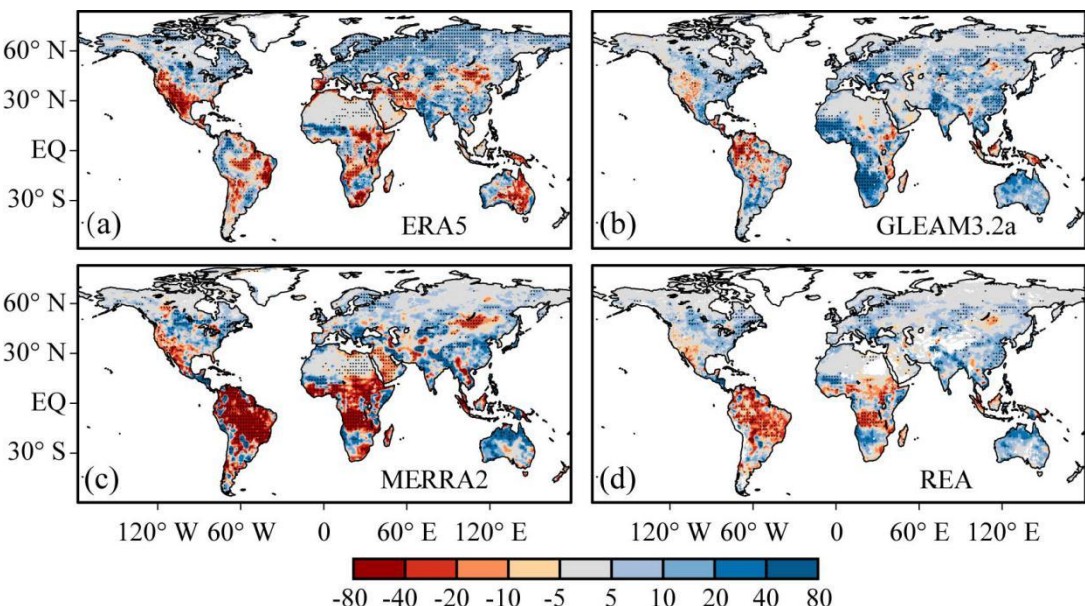

**Figure 9. Spatial distribution of linear trends of land evaporation for the period 1980-2017 (unit: mm decade-1). Stippling indicates statistically significant regions at 95% confidence level of the Mann-Kendall test.**

Figure 9 depicts the variation trends of multiple data sets during 1980-2017, where GLDAS-Noah2 has not been calculated for comparison due to two data sets including GlDAS-NOah2.0 and Gldas-Noah2.1 used throughout the period. Merged products shows that land evaporation significantly decreases in Amazon Plain in South America and Congo Basin in central Africa, while increases almost all over the world covering the east of North America, west of Europe, south of Asia and north of Oceania. The reduction trend in Amazon is observed in all data sets, with MERRA2 showing the most significant

and intense one. The decreasing trend in the Congo basin is detected in ERA5 and MERRA2, while an opposite one is observed in GLEAM3.2a. Burnett et al. (2020) found Congo basin becoming sunnier and less humid in recent years through the analysis of environmental data. In general, GLEAM is fairly close to land ET in tropical Africa (Schuttemeyer et al., 2007; Opoku-Duah et al., 2008; Andam-Akorful et al., 2015; Liu et al., 2016), while MERRA2 ET has the maximum temporal variability over Congo basin (Burnett et al., 2020; Crowhurst et al., 2020). The increase in the east of North

America, west of Europe and south of Asia is detectable in all data sets. The increasing trend in the north of Oceania is also detected in GLEAM3.2a and MERRA2, but not in ERA5.

Varying degrees of uncertainties exist in models based on satellites according to their theories, structural assumption and parameterization of the inputs. These limitations are mainly affected by changes in landscape, climatic and hydrological conditions (Xu et al., 2015). Changes in environmental conditions and extensive vegetation types in regional and global ET

estimation can lead to great uncertainties in ET products (Yilmaz et al., 2012; Liaqat and Choi, 2015; Liaqat et al., 2015; Khan et al., 2018). Apart from a few ET products, at present, the validation and analysis have been rarely adopted to reduce uncertainties. A few researches have made some attempts to reduce the uncertainties in hydrological applications

(Zhu et al., 2016), so far no such performance has been achieved yet. An emerging new technology REA method, which has the ability to combine different ET products (GLDAS, GLEAM, ERA5 and MERRA2) and is becoming increasingly

successful to resolve the issue of hydrological uncertainties and holds greater significance for in-depth assessment.

The uncertainties during the merging of ET products are driven by various factors including input errors, scaling effect and merging algorithm. Input errors are derived from single ET product and EC ground measurements. EC ground measurement determines the accuracy of the merged ET products, as it is considered to be the "true" value used to calibrate individual product, which persists an error of about 5-20%. They are usually found relatively accurate for ET acquisition (Foken et al.,

2006). The uncertainties from scaling effects are caused by the mismatch between the spatial resolution of the model and the tower footprint. The uncertainties of merging algorithms are caused by differential calculations of the weight of each product. In addition, there are other factors that lead to uncertainties. For example, uncertainties in remote sensing and meteorological data may affect the calculation of modeled ET. With regard to the model estimation, some common modeling assumptions such as estimation of potential evaporation, and shared inputs such as surface radiation make ET estimates from models

quite dependent, making the predicted errors correlated (Jimenez et al., 2017).

In general, reasonable information about hydrological variables on a global scale can be extracted from satellite- and reanalysis-based data sets, which is useful in regions lacking sufficient observations (Kim et al., 2018). For the discrepancy of the performances of reanalysis and satellite data under different mechanisms, the merging method can be used to verify the complementary strategy well to reflect the strengths of both. However, the complex structure of these merging methods

affects the efficiency of calculating the weight and limits their wider application. In contrast, the simplicity, computational efficiencies and reliable accuracy of REA method make it more preponderant when considering multi-source data sets (Giorgi & Mearns, 2002). On the basis of previous studies, this study focuses on the global performance of land ET merged product. To reduce the complexity, we introduced REA method to improve land ET estimation and merged three reanalysis data sets produced by separate algorithms. This method considers not only the performance of individual model, but also the

convergence of models involved in the merging process. Compared with individual products, REA merged ET product is found to outperform with significantly reduced root mean square error (RMSE = 0.05~ 0.27mm per day on average).

Compared with the widely used merging methods for land ET, REA has certain advantages. Specifically, Simple Average (SA) method is the simplest among all the methods, which depends on the assumption that the uncertainties are the same for each data set (Ershadi et al., 2014). However, this assumption lacks rationality in terms of the differences between data sets.

REA method gives corresponding weights according to the uncertainty of each data set, making up for this shorting. Empirical Orthogonal Function (EOF) based methods introduce biases due to little to no distinction between good and bad pixels in the reconstruction scheme (Feng et al., 2016). In addition, the complexity of the EOF method affects the calculation efficiency of the weights, resulting in high calculation cost. The REA method is easy to obtain the indicators used in the calculation of the weights, which greatly improves the calculation efficiency. Furthermore, there is a higher efficiency of the

REA method than the commonly used machine learning algorithm Support Vector Machine (SVM) when the sample size is

large (Yao et al., 2017a). Simple Taylor skill's Score (STS) method is highly dependent on the accuracy of the individual data, with the high demand of the quality of individual data set (Yao et al., 2017b). Not only the performance but also the convergence of the model does REA method depends on, making it less sensitive to the performance of individual data set. Since terrestrial ET is a complex variable coupled with energy, hydrology and carbon budget, it is difficult to accurately determine the optimal condition density function when using Bayesian Model Average (BMA) method (Yao et al., 2014; Zhu et al., 2016). Whereas, the two indicators adopted by the REA method, including the reliability and convergence of the model, are easy to obtain by the deviation between the simulated ET and the reference and the distance between the simulated ET and the ensemble average, respectively. Therefore, REA method possesses certain reliability and high efficiency in the terms of merging land ET.

However, REA method only takes into consideration the combination and relationship between each product and the reference data set. Although the improvement in the correlation coefficient is statistically significant, data sets both participating in the merging process and used as the reference have not been improved substantially. Therefore, the performance of REA method is highly dependent on the weight of individual data set, which is calibrated with the reference data. Consequently, the results also demonstrate that REA is more sensitive to higher qualities in GLEAM. As a result, where GLEAM has lower qualities, REA tends to have higher qualities. Meanwhile, they both have very similar qualities, or even higher in REA, at regions where GLEAM has higher correlations and lower differences with the insitu datasets (Fig S6). Thus REA is more sensitive to the reference data where it is more reliable. Future research needs to determine the physical mechanism of the inherent error of each product and strengthen the global quantification of ET products by combining the surface residual energy balance and water balance method without using any reference data set (Yao et al., 2017b; Baik et al., 2018).

## 4 Data availability

All data used in this study are freely available with the links given in section 2. A convenience copy of the merged global land evaporation product available at the time this paper was created has been registered with Zenodo and is available at https://doi.org/10.5281/zenodo.4595941 (Lu et al., 2021). Both daily and monthly dataset are provided.

## 5 Conclusion

In conclusion, we used CV as the indicator to select the merging regions with high data consistency, and the regions with low consistency were excluded from the merging scope, including the north of North America, west of South America, desert regions of mid-latitude Africa and Asia. We merged three land ET data sets, ERA5, GLDAS and MERRA2, respectively using REA method to generate a set of long sequence global daily ET data with a spatial resolution of 0.25 degree and a time span of 38 years. The quality of the merged product is found significantly improved when compared with

the three individual data sets selected for merging. Averaged global validations based on correlation coefficient and RMSE suggest that the merged product relatively has the best skill to capture ET dynamics over different locations and times among all data sets. Nonetheless, the results also indicate that no one product performs best across all ecosystems and timescales. Under different vegetation conditions when NDVI ranges from 0.34 to 0.75, the merged product can capture land evaporation dynamics most accurately.

The spatial distribution of the merged product was found to be highly consistent with other four data sets, indicating that the product successfully captures the spatial difference of land ET effectively. In terms of variation trends of global land ET, conclusions differ from numerous researches due to uncertainties of data used, and no agreement could be reached. Our merged product shows that there is a significant decreasing trend in Amazon plain in South America and Congo Basin in central Africa, and an increasing trend in the east of North America, west of Europe, south of Asia and north of Oceania. This is the first time we have used REA method more precisely in land ET data merging by avoiding the likely errors to be generated by physical mechanism.

Based on model weighting, the method of combining information into a new data set reflects the uncertainty behind climate change predictions, which should be explored in depth. A simple and flexible framework is provided for the exploration according to REA method. As we expected, the error was non-stationary. The weight of each day can be estimated by running a time window centered on the day for making the weight change over time. Shorter time windows produce more dynamic weights, nonetheless, the values may be noisier due to fewer samples available to estimate variability in the time series. Under the framework, cross-variable merging can be realized, that is, multiple related variables can be included for weight calculation (Xu et al., 2010). In future studies, the quality of merged products could be improved by adopting an appropriate time window to calculate the dynamic weight and considering more relevant variables to further reduce the uncertainty of the merged product.

**Author contributions.** JL conducted the research, completed the original draft, and revised it. TC, SL and DFTH contributed to data processing. GK, JP, TJ and BS revised the draft. GW, the corresponding author, contributed to conceptual designing, reviewing of the manuscript, funding acquisition, and project administration. All coauthors reviewed the manuscript and contributed to the writing process.

**Competing interests.** The authors declare that they have no conflict of interest.

**Acknowledgments.** This study is supported by National Key Research and Development Program of China (2017YFA0603701), National Natural Science Foundation of China (41875094) and the Postgraduate Research and Practice Innovation Program of Jiangsu Province (KYCX21_0957). Giri Kattel would like to acknowledge Longshan Professorship and the Talent Grant (1511582101011) in Nanjing University of Information Science and Technology (NUIST). We thank all contributors and also our external cooperation partners for their support and work. We thank Alexander Gruber and the two anonymous reviewers for the constructive and helpful comments which improved this paper substantially.

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
