# Peer review of "A Harmonized Global Land Evaporation Dataset from Model-based Products Covering 1980-2017"

_Earth System Science Data, 2021_

## Author Comment (AC1)

**Answer to Reviewer 1 ESSD-2021-61**

We thank Reviewer 1 for the comments. We provide here our responses to those comments and describe how we addressed them in the revised manuscript. The original reviewer comments are in normal black font while our answers appear in blue font. The corresponding edit in the manuscript are included in red font.

**General:**

This is a highly interesting and relevant dataset that might prove useful for numerous studies. Given that it claims to be a benchmark dataset more detail and discussion is necessary.

I would expect more detail on the validation part and the role of GLEAM as an independent reference dataset. The motivation for the latter needs to be discussed more honestly including the shortfalls of this approach.

A more detailed validation might reveal that the merged product does not necessarily outperform the other datasets at all sites/time periods (which is okay). At the moment only average global validation metrics are discussed.

**Response:**

Thank you for your very thoughtful comment and suggestions. We have added more detailed validations in different ecosystems and seasons, and have stated where and when the merged product does not necessarily outperform the individual product in the result section in the revised manuscript. The text there reads as: "*In the rest of the paper, the different land cover types listed, which are representative of different ecosystem types, will be simply referred to as ecosystems.*

*Table 2. The verification results including R and RMSD between daily Ground-measured ET and daily ET from different products in different ecosystems. Values in bold and blue indicates the highest quality while those in bold and red indicates the lowest.*

| Ecosystem type | ERA5 | | MERRA2 | | GLDAS | | REA | |
|---|---|---|---|---|---|---|---|---|
| | R | RMSD | R | RMSD | R | RMSD | R | RMSD |
| CRO | *0.66* | *1.24* | *0.55* | 1.42 | 0.60 | *1.48* | 0.60 | 1.38 |
| DBF | 0.76 | *1.06* | *0.71* | *1.23* | 0.74 | 1.19 | *0.77* | 1.07 |
| DNF | *0.81* | *0.55* | *0.73* | *0.75* | 0.77 | 0.73 | 0.80 | 0.62 |
| EBF | *0.72* | *1.08* | 0.61 | *1.59* | *0.58* | 1.36 | 0.65 | *1.08* |
| ENF | *0.66* | 1.03 | *0.66* | *1.21* | 0.67 | 1.05 | *0.73* | *0.88* |
| GRA | 0.72 | 1.05 | *0.77* | *1.11* | *0.70* | 1.09 | *0.77* | *0.94* |
| MF | 0.77 | *1.05* | *0.79* | *1.37* | *0.70* | 1.23 | 0.74 | 1.12 |
| OSH | *0.43* | *1.00* | 0.47 | 0.92 | 0.46 | 0.96 | *0.50* | *0.88* |
| SAV | *0.61* | 1.23 | 0.62 | *1.40* | 0.63 | 1.22 | *0.66* | *1.16* |
| WET | *0.57* | *1.40* | *0.44* | *1.66* | 0.47 | 1.56 | 0.46 | 1.59 |

| WSA | 0.68 | 1.24 | **0.63** | **1.46** | 0.64 | 1.17 | **0.70** | **1.13** |
|-----|------|------|----------|----------|------|------|----------|----------|

*Verification of ET products from different ecosystems has been conducted in order to further evaluate their performances. Table 2 describes quantitatively the performances of the ET products in 11 ecosystem types of site on a daily scale from two indicators, R and RMSD. The values in bold print and blue colour indicate the best performance of the four products and those in bold and red indicate the opposite. The results demonstrate that no individual product performs best across all ecosystems. For 42 ENF, 34 GRA, 9 OSH, 8 SAV and 6 WSA sites, REA has higher R and lower RMSD than individual products. For 23 DBF and 13 EBF sites, REA has a optimal R or RMSD; specifically with the highest R of 0.77 and second lowest RMSD of 1.07 mm per day for DBF, and the lowest RMSD of 1.08 mm per day and the second highest R of 0.65 for EBF. REA performs worse than at least one individual product at 57 other sites. Specifically, REA has lower R and higher RMSD than ERA5 at 17 CRO and 1 DNF sites. For 9 MF sites, REA has a lower R of 0.74 than ERA5 and MERRA2, and a higher RMSD of 1.12 mm per day than ERA5. For 19 WET sites, REA has a lower R of 0.46 and a higher RMSD of 1.59 mm per day than ERA5 and GLDAS. Generally, ERA5, MERRA2 and REA show the best performance respectively in five (including CRO, DNF, EBF and WET), two (including GRA and MF) and six (DBF, ENF, GRA, OSH, SAV and WSA) ecosystems in terms of R. Based on RMSD, both ERA5 and REA performed best in six ecosystems (with the former including CRO, DBF, DNF, EBF, MF and WET, and the latter including EBF, ENF, GRA, OSH, SAV and WSA). REA does not perform best across all ecosystems, however, it avoids the worst performance in any ecosystem. Taylor Diagram results of daily Ground-measured ET and ET from the different products in 11 ecosystems are put in support information (Fig. S1).*

**Table 3. The verification results including R and RMSD between monthly Ground-measured ET and monthly ET from different products in different ecosystems. Values in bold and blue indicates the highest quality while those in bold and red indicates the lowest.**

| Ecosystem type | ERA5 | | MERRA2 | | GLDAS | | REA | |
|----------------|------|------|--------|------|-------|------|------|------|
| | R | RMSD | R | RMSD | R | RMSD | R | RMSD |
| CRO | 0.71 | 31.84 | **0.58** | 38.33 | 0.64 | **39.59** | **0.73** | **31.14** |
| DBF | 0.84 | 26.61 | **0.77** | **32.40** | 0.83 | 30.27 | **0.86** | **26.17** |
| DNF | **0.93** | **10.62** | **0.84** | 18.63 | 0.91 | **18.96** | 0.91 | 11.01 |
| EBF | **0.81** | **25.85** | 0.71 | **40.32** | **0.69** | 32.21 | 0.78 | 27.06 |
| ENF | 0.74 | 25.01 | **0.72** | **31.69** | **0.76** | 25.38 | **0.76** | **23.91** |
| GRA | 0.77 | 27.11 | **0.83** | 28.59 | 0.77 | **26.54** | 0.77 | 27.53 |
| MF | 0.83 | 27.57 | **0.85** | **37.98** | **0.75** | 32.90 | 0.83 | **26.99** |
| OSH | **0.45** | **25.34** | 0.51 | **23.39** | 0.52 | 23.75 | **0.53** | 24.54 |
| SAV | **0.65** | 32.05 | **0.65** | **38.08** | 0.67 | 31.88 | **0.68** | **31.45** |
| WET | 0.61 | 38.30 | **0.46** | **46.92** | 0.49 | 43.78 | **0.64** | **37.35** |
| WSA | **0.73** | 33.71 | **0.68** | **38.54** | 0.72 | **28.62** | **0.73** | 34.49 |

*Similar to Table 2, Table 3 shows the performance of ET products in different ecosystems on a monthly scale. Compared with daily scale, the performance of each product has changed, among which all of the R becomes higher. REA has higher R and lower RMSD than individual products at 17 CRO, 23 DBF, 42 ENF, 8 SAV and 19 WET sites. It has an optimal R or RMSD at 9 MF, 9 OSH and 6 WSA sites. At other 58 sites, REA has a worse performance than at least one individual product. For 1 DNF and 13 EBF sites, REA has lower R and higher RMSD than ERA5. For 34 GRA sites, ERA5 has a lower R of 0.77 than MERRA2, and a higher RMSD of 27.53 mm per month than ERA5 and GLDAS. Similar to the daily scale, REA does not have a better performance than any individual product in all ecosystems, but is superior to at least one individual one. Similarly, the Taylor charts of monthly Ground-measured ET and ET from the different products in 11 ecosystems are put in support information (Fig. S2).*

**Table 4. The verification results including R and RMSD between daily Ground-measured ET and daily ET from different products in different seasons. Values in bold and blue indicates the highest quality while those in bold and red indicates the lowest.**

| Ecosystem type | ERA5 | | MERRA2 | | GLDAS | | REA | |
|---|---|---|---|---|---|---|---|---|
| | R | RMSD | R | RMSD | R | RMSD | R | RMSD |
| MAM | **0.63** | **1.12** | 0.62 | **1.44** | **0.58** | 1.21 | **0.63** | 1.15 |
| JJA | **0.44** | **1.45** | **0.41** | **1.77** | 0.42 | 1.69 | **0.44** | 1.47 |
| SON | 0.64 | **0.96** | 0.61 | 0.93 | **0.60** | 0.93 | **0.65** | **0.84** |
| DJF | 0.78 | 0.77 | 0.75 | **0.83** | **0.74** | 0.77 | **0.79** | **0.70** |

*In addition to different ecosystems, seasonal validation has been carried out to try to find out how each ET product performs in different seasons. Table 4 shows the performance of four ET products in different seasons on a daily scale. In general, REA has a better performance than individual products in autumn and winter, while in spring and summer it has a worse performance than ERA5. The R of REA for all seasons varies from 0.44 to 0.79, which remains optimal. In spring and summer, its R is as high as ERA5, and RMSD is second only to the best-performing ERA5. In addition, MERRA2 and GLDAS perform similarly. Specifically, the R of MERRA2 is higher than that of GLDAS in all seasons except summer, however, RMSD of GLDAS is lower than that of MERRA2. In terms of the whole year, the R of all products is higher in winter and lower in summer than other seasons. Similarly, RMSD is the highest in summer and the lowest in winter, which is mainly caused by the large variation and absolute value of land ET in summer. The Taylor charts of daily Ground-measure ET and ET from the different products in four seasons are put in support information (Fig. S3).*

**Table 5. The verification results including R and RMSD between monthly Ground-measured ET and monthly ET from different products in different seasons. Values in bold and blue indicates the highest quality while those in bold and red indicates the lowest.**

| Ecosystem type | ERA5 | | MERRA2 | | GLDAS | | REA | |
|---|---|---|---|---|---|---|---|---|
| | R | RMSD | R | RMSD | R | RMSD | R | RMSD |
| MAM | 0.69 | 28.93 | 0.68 | *39.04* | *0.65* | 30.54 | *0.71* | *28.51* |
| JJA | 0.42 | 37.67 | *0.38* | *48.15* | 0.41 | 44.89 | *0.45* | *36.71* |
| SON | 0.71 | *23.66* | 0.67 | 23.08 | *0.66* | *22.58* | *0.72* | 23.59 |
| DJF | *0.85* | 18.18 | *0.84* | *19.24* | 0.84 | *17.14* | *0.85* | 18.35 |

*Compared with the daily scale, the performance of REA varies greatly in different seasons at the monthly scale (Table 5). In spring and summer, REA performs better than all individual products. While in autumn and winter, REA has a higher R of 0.72 and 0.85 than individual products, and slightly larger RMSD of 23.59 and 18.35 mm per month than MERRA2 and GLDAS. Like the daily scale, the performance of all products is still better in winter and worse in summer. Although the performances of REA at the daily and monthly scales vary in each season, the R is always higher than individual products, indicating the highly consistent of variation of REA with the observations. As well, the Taylor charts of monthly Ground-measure ET and ET from the different products in four seasons are put in support information (Fig. S4).".*

We have made a supplement in the conclusion section in the revised manuscript. The text there reads as: "*Averaged global validations based on correlation coefficient and RMSE suggest that the merged product relatively has the best skill to capture ET dynamics over different locations and times among all data sets. Nonetheless, the results also indicate that no one product performs best across all ecosystems and timescales. Under different vegetation conditions when NDVI ranges from 0.34 to 0.75, the merged product can capture land evaporation dynamics most accurately.*".

We have added the clarification of the role of GLEAM as an independent reference data set in the data section. GLEAM is a long sequence data set predominantly based on remote sensing observations, and on occasion, reanalysis data. GLEAM is unlike traditional land models, such as found in ERA5, MERRA2 and GLDAS, in that it is driven by satellite observations to obtain evaporation estimates. The version of GLEAM here relies very little on reanalysis datasets (only radiation and temperature of ERA-Interim). Therefore, GLEAM has the most independence relative to the model-based products.

We have added the description of the relative independence of GLEAM in the revised manuscript. The text there reads as: "*In addition, GLEAM is a long sequence data set predominantly based on remote sensing observations, and on occasion, reanalysis data. GLEAM is unlike traditional land models, such as found in ERA5, MERRA2 and GLDAS, in that it is driven by satellite observations to obtain evaporation estimates. The version of GLEAM here relies very little on reanalysis datasets (only radiation and temperature of ERA-Interim). Therefore, GLEAM has the most independence relative to the model-based products, which is selected as the reference data due to its relative independence.*".

**Language:**

Please revise the language before publication, best with a native speaker, e.g. "Land evaporation (ET) plays a crucial role in the hydrological and energy cycle.", the first sentence of the abstract. Next sentence: What are numerical products? The lack of …

"were distributed across the east-west direction banding manner" hard to understand.

Please also check the language of the data described on the data portal.

**Response:**

Thank you for your cogent advice. We have revised the language carefully, including but not limited to the following details.

We have changed "numerical products" to "model-based products" in the revised manuscript. The text there reads as: "*However, the widely used model-based products are still subject to uncertainties due to different model parameterizations and forcing datasets.*".

We have changed "lack of" to "the lack of" in the revised manuscript. The text there reads as: "*The lack of available observed data has further complicated the estimation.*".

We have changed "were distributed across the east-west direction banding manner" to "were zonally distributed along the east-west direction" in the revised manuscript. The text there reads as: "*The results showed that the merged product performed well under a variety of vegetation cover conditions as the weights were zonally distributed along the east-west direction, with greater differences near the equator.*".

**Major:**

Please add a lot more details on the validation across different Fluxnet sites, ecosystems, years etc. How does data availability of in situ data in different years affect the validation? Can a significance test be included? Where (e.g. ecosystems, latitude, longitude) / When does the merged product perform better than the individual model output, where/when worse or similar?

**Response:**

Thank you for your very thoughtful comment. We have added details on validation. First of all, we have labeled the ecosystem of sites in Fig. 1 and added the description in the revised manuscript. The text there reads as: "

[Figure]

**Figure 1. a) Spatial distribution of 181 in-situ flux EC sites across the world and b) The number of sites for different time span.".**

*The flux towers are located in 11 land cover types including 17 Croplands (CRO), 23 Deciduous Broadleaf Forest (DBF), 1 Deciduous Needleleaf Forest (DNF), 13 Evergreen Broadleaf Forest (EBF), 42 Evergreen Needleleaf Forest (ENF), 34 Grasslands (GRA), 9 Mixed Forest (MF), 9 Open Shrublands (OSH), 8 Savannas (SAV), 19 Permanent Wetlands (WET) and 6 Woody Savannas (WSA) sites. In the rest of the paper, the different land cover types listed, which are representative of different ecosystem types, will be simply referred to as ecosystems.".*

Afterwards, we have added the validation of individual and merged ET products in different ecosystems and seasons, and have stated where and when the merged product has a better or worse performance than individual product in the revised manuscript. The text there reads as: "

**Table 2. The verification results including R and RMSD between daily Ground-measured ET and daily ET from different products in different ecosystems. Values in bold and blue indicates the highest quality while those in bold and red indicates the lowest.**

| Ecosystem type | ERA5 | | MERRA2 | | GLDAS | | REA | |
|---|---|---|---|---|---|---|---|---|
| | R | RMSD | R | RMSD | R | RMSD | R | RMSD |

| | ERA5 | | MERRA2 | | GLDAS | | REA | |
|---|---|---|---|---|---|---|---|---|
| | R | RMSD | R | RMSD | R | RMSD | R | RMSD |
| CRO | *0.66* | *1.24* | *0.55* | 1.42 | 0.60 | *1.48* | 0.60 | 1.38 |
| DBF | 0.76 | *1.06* | *0.71* | *1.23* | 0.74 | 1.19 | *0.77* | 1.07 |
| DNF | *0.81* | *0.55* | *0.73* | *0.75* | 0.77 | 0.73 | 0.80 | 0.62 |
| EBF | *0.72* | *1.08* | 0.61 | *1.59* | *0.58* | 1.36 | 0.65 | *1.08* |
| ENF | *0.66* | 1.03 | *0.66* | *1.21* | 0.67 | 1.05 | *0.73* | *0.88* |
| GRA | 0.72 | 1.05 | *0.77* | *1.11* | *0.70* | 1.09 | *0.77* | *0.94* |
| MF | 0.77 | *1.05* | *0.79* | *1.37* | *0.70* | 1.23 | 0.74 | 1.12 |
| OSH | *0.43* | *1.00* | 0.47 | 0.92 | 0.46 | 0.96 | *0.50* | *0.88* |
| SAV | *0.61* | 1.23 | 0.62 | *1.40* | 0.63 | 1.22 | *0.66* | *1.16* |
| WET | *0.57* | *1.40* | *0.44* | *1.66* | 0.47 | 1.56 | 0.46 | 1.59 |
| WSA | 0.68 | 1.24 | *0.63* | *1.46* | 0.64 | 1.17 | *0.70* | *1.13* |

*Verification of ET products from different ecosystems has been conducted in order to further evaluate their performances. Table 2 describes quantitatively the performances of the ET products in 11 ecosystem types of site on a daily scale from two indicators, R and RMSD. The values in bold print and blue colour indicate the best performance of the four products and those in bold and red indicate the opposite. The results demonstrate that no individual product performs best across all ecosystems. For 42 ENF, 34 GRA, 9 OSH, 8 SAV and 6 WSA sites, REA has higher R and lower RMSD than individual products. For 23 DBF and 13 EBF sites, REA has a optimal R or RMSD; specifically with the highest R of 0.77 and second lowest RMSD of 1.07 mm per day for DBF, and the lowest RMSD of 1.08 mm per day and the second highest R of 0.65 for EBF. REA performs worse than at least one individual product at 57 other sites. Specifically, REA has lower R and higher RMSD than ERA5 at 17 CRO and 1 DNF sites. For 9 MF sites, REA has a lower R of 0.74 than ERA5 and MERRA2, and a higher RMSD of 1.12 mm per day than ERA5. For 19 WET sites, REA has a lower R of 0.46 and a higher RMSD of 1.59 mm per day than ERA5 and GLDAS. Generally, ERA5, MERRA2 and REA show the best performance respectively in five (including CRO, DNF, EBF and WET), two (including GRA and MF) and six (DBF, ENF, GRA, OSH, SAV and WSA) ecosystems in terms of R. Based on RMSD, both ERA5 and REA performed best in six ecosystems (with the former including CRO, DBF, DNF, EBF, MF and WET, and the latter including EBF, ENF, GRA, OSH, SAV and WSA). REA does not perform best across all ecosystems, however, it avoids the worst performance in any ecosystem. Taylor Diagram results of daily Ground-measured ET and ET from the different products in 11 ecosystems are put in support information (Fig. S1).*

**Table 3. The verification results including R and RMSD between monthly Ground-measured ET and monthly ET from different products in different ecosystems. Values in bold and blue indicates the highest quality while those in bold and red indicates the lowest.**

| Ecosystem type | ERA5 | | MERRA2 | | GLDAS | | REA | |
|---|---|---|---|---|---|---|---|---|
| | R | RMSD | R | RMSD | R | RMSD | R | RMSD |
| CRO | 0.71 | 31.84 | *0.58* | 38.33 | 0.64 | *39.59* | *0.73* | *31.14* |

| | ERA5 R | ERA5 RMSD | MERRA2 R | MERRA2 RMSD | GLDAS R | GLDAS RMSD | REA R | REA RMSD |
|---|---|---|---|---|---|---|---|---|
| DBF | 0.84 | 26.61 | *0.77* | *32.40* | 0.83 | 30.27 | *0.86* | *26.17* |
| DNF | *0.93* | *10.62* | *0.84* | 18.63 | 0.91 | *18.96* | 0.91 | 11.01 |
| EBF | *0.81* | *25.85* | 0.71 | *40.32* | *0.69* | 32.21 | 0.78 | 27.06 |
| ENF | 0.74 | 25.01 | *0.72* | *31.69* | *0.76* | 25.38 | *0.76* | *23.91* |
| GRA | 0.77 | 27.11 | *0.83* | 28.59 | 0.77 | *26.54* | 0.77 | 27.53 |
| MF | 0.83 | 27.57 | *0.85* | *37.98* | *0.75* | 32.90 | 0.83 | *26.99* |
| OSH | *0.45* | *25.34* | 0.51 | *23.39* | 0.52 | 23.75 | *0.53* | 24.54 |
| SAV | *0.65* | 32.05 | *0.65* | *38.08* | 0.67 | 31.88 | *0.68* | *31.45* |
| WET | 0.61 | 38.30 | *0.46* | *46.92* | 0.49 | 43.78 | *0.64* | *37.35* |
| WSA | *0.73* | 33.71 | *0.68* | *38.54* | 0.72 | *28.62* | *0.73* | 34.49 |

*Similar to Table 2, Table 3 shows the performance of ET products in different ecosystems on a monthly scale. Compared with daily scale, the performance of each product has changed, among which all of the R becomes higher. REA has higher R and lower RMSD than individual products at 17 CRO, 23 DBF, 42 ENF, 8 SAV and 19 WET sites. It has an optimal R or RMSD at 9 MF, 9 OSH and 6 WSA sites. At other 58 sites, REA has a worse performance than at least one individual product. For 1 DNF and 13 EBF sites, REA has lower R and higher RMSD than ERA5. For 34 GRA sites, ERA5 has a lower R of 0.77 than MERRA2, and a higher RMSD of 27.53 mm per month than ERA5 and GLDAS. Similar to the daily scale, REA does not have a better performance than any individual product in all ecosystems, but is superior to at least one individual one. Similarly, the Taylor charts of monthly Ground-measured ET and ET from the different products in 11 ecosystems are put in support information (Fig. S2).*

**Table 4. The verification results including R and RMSD between daily Ground-measured ET and daily ET from different products in different seasons. Values in bold and blue indicates the highest quality while those in bold and red indicates the lowest.**

| Ecosystem type | ERA5 R | ERA5 RMSD | MERRA2 R | MERRA2 RMSD | GLDAS R | GLDAS RMSD | REA R | REA RMSD |
|---|---|---|---|---|---|---|---|---|
| MAM | *0.63* | *1.12* | 0.62 | *1.44* | *0.58* | 1.21 | *0.63* | 1.15 |
| JJA | *0.44* | *1.45* | *0.41* | *1.77* | 0.42 | 1.69 | *0.44* | 1.47 |
| SON | 0.64 | *0.96* | 0.61 | 0.93 | *0.60* | 0.93 | *0.65* | *0.84* |
| DJF | 0.78 | 0.77 | 0.75 | *0.83* | *0.74* | 0.77 | *0.79* | *0.70* |

*In addition to different ecosystems, seasonal validation has been carried out to try to find out how each ET product performs in different seasons. Table 4 shows the performance of four ET products in different seasons on a daily scale. In general, REA has a better performance than individual products in autumn and winter, while in spring and summer it has a worse performance than ERA5. The R of REA for all seasons varies from 0.44 to 0.79, which remains optimal. In spring and summer, its R is as high as ERA5, and RMSD is second only to the best-performing ERA5. In addition, MERRA2 and GLDAS perform similarly. Specifically, the R of MERRA2 is higher than that of GLDAS in all seasons except summer, however, RMSD of GLDAS*

*is lower than that of MERRA2. In terms of the whole year, the R of all products is higher in winter and lower in summer than other seasons. Similarly, RMSD is the highest in summer and the lowest in winter, which is mainly caused by the large variation and absolute value of land ET in summer. The Taylor charts of daily Ground-measure ET and ET from the different products in four seasons are put in support information (Fig. S3).*

*Table 5. The verification results including R and RMSD between monthly Ground-measured ET and monthly ET from different products in different seasons. Values in bold and blue indicates the highest quality while those in bold and red indicates the lowest.*

| Ecosystem type | ERA5 | | MERRA2 | | GLDAS | | REA | |
|---|---|---|---|---|---|---|---|---|
| | R | RMSD | R | RMSD | R | RMSD | R | RMSD |
| MAM | 0.69 | 28.93 | 0.68 | **39.04** | **0.65** | 30.54 | **0.71** | **28.51** |
| JJA | 0.42 | 37.67 | **0.38** | **48.15** | 0.41 | 44.89 | **0.45** | **36.71** |
| SON | 0.71 | **23.66** | 0.67 | 23.08 | **0.66** | **22.58** | **0.72** | 23.59 |
| DJF | **0.85** | 18.18 | **0.84** | **19.24** | 0.84 | **17.14** | **0.85** | 18.35 |

*Compared with the daily scale, the performance of REA varies greatly in different seasons at the monthly scale (Table 5). In spring and summer, REA performs better than all individual products. While in autumn and winter, REA has a higher R of 0.72 and 0.85 than individual products, and slightly larger RMSD of 23.59 and 18.35 mm per month than MERRA2 and GLDAS. Like the daily scale, the performance of all products is still better in winter and worse in summer. Although the performances of REA at the daily and monthly scales vary in each season, the R is always higher than individual products, indicating the highly consistent of variation of REA with the observations. As well, the Taylor charts of monthly Ground-measure ET and ET from the different products in four seasons are put in support information (Fig. S4).".*

Figure 1 shows that the data availability of in situ data occurs over different periods. Ideally, a fair evaluation of the products could be done if the availability of the datasets fully overlapped. However, here, the limited available overlapping times of the insitu datasets makes their use quite inconsistence. Nonetheless, they are still useful since they offer an objective evaluation source. As such no filtering was applied to select specific data apart from the quality control applied. We have added the description in the revised manuscript. The text there reads as: "*Figure 1 shows that the data availability of in situ data occurs over different periods. The data cover the period of 1992-2014, including at least 1 year reliable data. As shown in Fig. 1b, the periods vary from 1 to 19 years, with 14, 32, 32, 13 and 9 sites reporting 1, 2, 3, 4 and 5 years of data respectively, accounting for 55 percent of the sites. Ideally, a fair evaluation of the products could be done if the availability of the datasets fully overlapped. However, here, the limited available overlapping times of the insitu datasets makes their use quite inconsistence. Nonetheless, they are still useful since*

*they offer an objective evaluation source. As such no filtering was applied to select specific data apart from the quality control applied.*".

We have made a supplement in the conclusion section in the revised manuscript. The text there reads as: "*Averaged global validations based on correlation coefficient and RMSE suggest that the merged product relatively has the best skill to capture ET dynamics over different locations and times among all data sets. Nonetheless, the results also indicate that no one product performs best across all ecosystems and timescales. Under different vegetation conditions when NDVI ranges from 0.34 to 0.75, the merged product can capture land evaporation dynamics most accurately.*".

[Figure]

***Figure S1. Taylor diagrams of daily Ground-measure ET and ET from different products in different ecosystems.***

[Figure]

*Figure S2. Taylor diagrams of monthly Ground-measure ET and ET from the different products in different ecosystems.*

[Figure]

*Figure S3. Taylor diagrams of daily Ground-measured ET and ET from different products in different seasons.*

[Figure]

*Figure S4. Taylor diagrams of monthly Ground-measured ET and ET from different products in different seasons.*

The role of GLEAM needs to be clarified! What is the motivation of using a fourth model as an independent reference dataset? What is the influence of GLEAM on the final product? The weight map is I assume largely influenced by how the three models agree with GLEAM, but why is this objective? Could the merging have been done using in-situ data to determine which model performs best too?

**Response:**

Thank you very much for your comment. GLEAM serves as the reference data which is required in this approach for the merging process. There is a further requirement of the REA method that the reference data be independent of the products participating in the merging process to avoid cross-correlation errors.

According to the first reliability criterion of the REA method (model performance), GLEAM affects the weights for the individual products used to calculate the final product by influencing the first indicator $R_B$, a factor to measure the reliability of the model by deviation between individual products and reference.

Indeed, the weight map is influenced by how the three individual products agree with the reference data GLEAM.

GLEAM rather than in-situ data was chosen as the reference data because in-situ data exists in very limited regions, whereas GLEAM is grid data with a spatial resolution of 0.25 degrees. In addition, GLEAM is the long sequence data set based on remote sensing observations with the advantage of the maximum use of remote sensing observations. Reanalysis data of very limited number of variables (only radiation and temperature) are used. Therefore, it is selected as the reference data due to its relative independence.

We have enriched the reasons for choosing GLEAM as the reference data in the revised manuscript. The text there reads as: "*Ideally, in-situ data would be the first choice to be used as the reference data for the merging. However, these point-scale datasets are very scarce globally and only representative of their immediate locations. Therefore, area-averaged grid-scale estimates offer a better alternative. Additionally, GLEAM is not a traditional terrestrial model as found in ERA5, MERRA2 and GLDAS, but outputs evaporation estimates driven by satellite observations and relies on only radiation and temperature inputs from reanalysis products. As such GLEAM offers a higher level of independence than the other products are essentially traditional models corrected with observations through data assimilation.*".

GLEAM (a) actually uses ERA-Interim as input and will therefore be highly correlated with ERA5, also all reanalysis products will be quite correlated and are definitely not independent as they will assimilate millions of the same observations even if the models are different.

**Response:**

Thank you for that comment. GLEAM is a long sequence data set predominantly based on remote sensing observations, and on occasion, reanalysis data. GLEAM is unlike traditional land models, such as found in ERA5, MERRA2 and GLDAS, in that it is driven by satellite observations to obtain evaporation estimates. The version of GLEAM here relies very little on reanalysis datasets (only radiation and temperature of ERA-Interim). Although the "b" version is based entirely on satellite observations, its length of time period is too short to achieve the purpose. Therefore, GLEAM has the most independence relative to the model-based products. GLEAM would be expected to correlate with ERA5 to a certain degree that would warrant thorough analysis to actually quantify. This is because only the radiation and temperature inputs are from reanalysis products, in this case, ERA Interim, which has been noted in several publications to have some appreciable differences with ERA5 which comes with several updates and changes from its predecessor. Secondly, since GLEAM is predominantly driven by satellite observations, it is expected that the impacts of the

reanalysis inputs would not have a direct impact on the output of the merging process. Given these reasons, we believe that ERA5 would reliably be useful in the study, especially over areas significant uncertainties may be found in the reference.

The model based products do employ several satellite inputs through data assimilation to update the prognostic states of the models. MERRA2 and ERA5 are based on brightness temperatures that are assimilated into their atmospheric models, and only indirectly impact the land states. Additionally, the climate models, as well as data assimilations schemes of the two are significantly different. Thus, we can expect some clear differences as a result of the different parameterization schemes. GLDAS, on the other hand, is not a reanalysis product, but uses updated atmospheric forcings from an earlier run which are used to drive the land model to obtain its outputs. Therefore, it does hold some independence from the other two. Eventually, the aim is that the different strengths from these models would be leveraged in the combination. Nonetheless, we do expect some cross-correlation errors to a certain degree.

We have added the description of the relative independence of GLEAM in the revised manuscript. The text there reads as: "*In addition, GLEAM is a long sequence data set predominantly based on remote sensing observations, and on occasion, reanalysis data. GLEAM is unlike traditional land models, such as found in ERA5, MERRA2 and GLDAS, in that it is driven by satellite observations to obtain evaporation estimates. The version of GLEAM here relies very little on reanalysis datasets (only radiation and temperature of ERA-Interim). Therefore, GLEAM has the most independence relative to the model-based products, which is selected as the reference data due to its relative independence.*".

Further, there is no requirement for the independence of the data involved in the merging process in the REA method, in consequence the correlation between the three reanalysis products including ERA5, MERRA2 and GLDAS does not affect the final product.

Again, this really needs to be clarified in the manuscript. One might argue that the resulting dataset uses the well-performing GLEAM dataset as a reference to compute the weights, however, one can not argue at all that GLEAM will be correct, or even superior to the other datasets across all pixels/time periods!

**Response:**

We share the reviewer's concern as well which was discussed at length by the authors during the study's design.

Firstly, GLEAM was included in preliminary evaluations of the ET estimates. By including GLEAM in the evaluations, we aim to assess regions where the reference

data will be potentially less reliable. This also provides the information on regions of high uncertainties with respect to the reference data, which becomes very useful with applications. Thus, we obtained a good understanding of the skill of GLEAM prior to its use as the reference data. Secondly, an aim of the study is to leverage the uniqueness of GLEAM (as discussed above) to combine the model-based products. It is expected that GLEAM's over-reliance on observations states would serve as some sort of benchmark to estimate the weights of the model-based products. Thus, the goal is not based on a superior skill of GLEAM but its added value due to its uniqueness relative to the model-based products, which we believe, does have merits.

We have added the clarification of the role of GLEAM in the revised manuscript. The text there reads as: "*Ideally, in-situ data would be the first choice to be used as the reference data for the merging. However, these point-scale datasets are very scarce globally and only representative of their immediate locations. Therefore, area-averaged grid-scale estimates offer a better alternative. Additionally, GLEAM is not a traditional terrestrial model as found in ERA5, MERRA2 and GLDAS, but outputs evaporation estimates driven by satellite observations and relies on only radiation and temperature inputs from reanalysis products. As such GLEAM offers a higher level of independence than the other products are essentially traditional models corrected with observations through data assimilation.*".

How in the end does this method compare to other merging methods, at least qualitatively? Add this to the discussion/conclusions.

**Response:**

Thank you for your very thoughtful comment. We have added the comparison to the discussion section in the revised manuscript. The text there reads as: "*Compared with the widely used merging methods for land ET, REA has certain advantages. Specifically, Simple Average (SA) method is the simplest among all the methods, which depends on the assumption that the uncertainties are the same for each data set (Ershadi et al., 2014). However, this assumption lacks rationality in terms of the differences between data sets. REA method gives corresponding weights according to the uncertainty of each data set, making up for this shorting. Empirical Orthogonal Function (EOF) based methods introduce biases due to little to no distinction between good and bad pixels in the reconstruction scheme (Feng et al., 2016). In addition, the complexity of the EOF method affects the calculation efficiency of the weights, resulting in high calculation cost. The REA method is easy to obtain the indicators used in the calculation of the weights, which greatly improves the calculation efficiency. Furthermore, there is a higher efficiency of the REA method than the commonly used machine learning algorithm Support Vector Machine (SVM) when the*

*sample size is large (Yao et al., 2017a). Simple Taylor skill's Score (STS) method is highly dependent on the accuracy of the individual data, with the high demand of the quality of individual data set (Yao et al., 2017b). Not only the performance but also the convergence of the model does REA method depends on, making it less sensitive to the performance of individual data set. Since terrestrial ET is a complex variable coupled with energy, hydrology and carbon budget, it is difficult to accurately determine the optimal condition density function when using Bayesian Model Average (BMA) method (Yao et al., 2014; Zhu et al., 2016). Whereas, the two indicators adopted by the REA method, including the reliability and convergence of the model, are easy to obtain by the deviation between the simulated ET and the reference and the distance between the simulated ET and the ensemble average, respectively. Therefore, REA method possesses certain reliability and high efficiency in the terms of merging land ET.".*

The conclusion interestingly mentions the temporal nature of the errors, this is however not really mentioned in the main part unless I missed it. This is a very interesting aspect.

**Response:**

Thank you for that comment. It refers to an extension of the REA method. The basic version of the REA method used in this study generated weights at each grid, while the weights at different points in time is unacquirable. However, the performance of a data set varies across all time points in a certain region, with some time performing better and some worse. The advanced version of the REA method generates merged data in all time points by sliding window, making the merged data perform as optimally as possible at all points in time, which is planned to be implemented in the future work. Data of fixed length are obtained by setting the sliding window. The REA method generates the weight and further the merged data of a point in time. As the sliding window moves, merged data at different points in time are generated.

We have added relevant descriptions in the method section in the revised manuscript. The text there reads as: "*However, the performance of a data set varies across all time points in a certain region, with some time performing better and some worse. The advanced version of the REA method generates merged data in all time points by sliding window, making the merged data perform as optimally as possible at all points in time, which is planned to be implemented in the future work. Data of fixed length are obtained by setting the sliding window. The REA method generates the weight and further the merged data of a point in time. As the sliding window moves, merged data at different points in time are generated.*".

**Minor:**

L41: "from in-situ observations and satellite inversion" Which inverse satellite retrievals exist for evaporation? I'm wasn't aware of any and am curious. Add examples/citations.

**Response:**

Thank you for pointing this out. Land evaporation obtained from satellite inversion includes MOD16, GLEAM and SSEBop, etc. MOD16 (Mu et al., 2007) (MODIS Global Evapotranspiration Project) is retrieved from MODIS remote sensing data, meteorological reanalysis data and improved P-M formula. GLEAM (Miralles et al., 2011a) (Global Land-Surface Evaporation: The Amsterdam Methodology) is derived from the inversion of multi-source remote sensing data, meteorological reanalysis data and the improved Priestley-Taylor (P-T) formula. SSEBop (Senay et al., 2013) (Operational Simplified Surface Energy Balance model) is retrieved based on MODIS remote sensing data, meteorological reanalysis data and simplified surface energy balance equation. We have added the examples and the corresponding citations in the revised manuscript. The text there reads as: "*In recent years, multiple land evaporation data sets at global scales are also available from in situ observations and satellite inversion, such as MOD16 (Mu et al., 2007), GLEAM (Miralles et al., 2011a) and SSEBop (Senay et al., 2013), etc.*".

Reference:

Mu, Q., Heinsch, F. A., Zhao, M., and Running, S. W.: Development of a Global Evapotranspiration Algorithm based on MODIS and Global Meteorology Data, Remote Sens. Environ., 111, 519-536, http://doi.org/10.1016/j.rse.2007.04.015, 2007.

Miralles, D. G., Holmes, T. R. H., De Jeu, R. A. M., Gash, J. H., Meesters, A. G. C. A., Dolman, A. J.: Global Land-Surface Evaporation Estimated from Satellite-based Observations, Hydrol. Earth Syst. Sc., 15, 453-469, http://doi.org/10.5194/hess-15-453-2011, 2011a.

Senay, G. B., Bohms, S., Singh, R. K., Gowda, P. H., Velpuri, N. M., Alemu, H., Verdin, J. P.: Operational Evapotranspiration Mapping Using Remote Sensing and Weather Datasets: A New Parameterization for the SSEB Approach, J. Am. Water Resour. As., 49, 577-591, http://doi.org/10.1111/jawr.12057, 2013.

L48: Can there ever be a global benchmark dataset if there is no in-situ data available everywhere? In the end, any dataset will be a proxy in these areas.

**Response:**

Thank you for your cogent advice. We used an inappropriate description of our data. We share this view and have changed "benchmark" to "proxy" or deleted the

descriptions of "benchmark" in the revised manuscript.

The text there reads as: "*On account of these reasons, an alternative product is developed which leverages the strengths of widely used existing model-based ET products.*".

"*Hence, there is an urgency to define the global proxy land ET with lower uncertainties for climate-induced hydrology and energy change.*".

"*To summarize, the uncertainty of land evaporation estimation will introduce adverse errors in various aspects, which puts the global proxy ET data set with lower uncertainties in a crucial position.*".

"*To reduce the complexity, we introduced REA method to improve land ET estimation and merged three reanalysis data sets produced by separate algorithms.*".

L62: "However, the practical application of maximized R method is usually found limited due to its use of only two most relevant in given data sources." I couldn't understand this sentence

**Response:**

Thank you for pointing this out. In given multiple data, only two sets of data that are most relevant to the reference data are selected to participate in the calculation, resulting the practical application of maximized R method limited. We have modified this sentence in the revised manuscript. The text there reads as: "*However, the practical application of maximized R method is usually found limited because only two sets from the given multiple data that are most relevant to the reference data are selected to participate in the calculation.*".

Can paragraphs from 51 and 58 be combined? They seem to partly repeat the same issue.

**Response:**

Thank you very much for your comment. We have combined the two paragraphs.

L64: Is MSWEP a good example of a merging method? It's the one I read about the most, but I'm not an expert in precipitation merging.

**Response:**

Thank you for your very thoughtful comment. MSWEP is a good example of a merging method. In the global validation, the quality of MSWEP is better than that of

the four most advanced gauge-adjusted precipitation data sets (Beck et al., 2017), which proves the effectiveness of the method. Global comparison shows that MSWEP V2 presents a more realistic spatial pattern in mean, magnitude and frequency (Beck et al., 2019).

We have added the method in the revised manuscript. The text there reads as: "*In regard to precipitation data merging, geographically weighted regression algorithm (Xu et al., 2015), conditional merging (Baik et al., 2016), geographical difference analysis (Cheema and Bastiaanssen, 2012), geographic ratio analysis (Duan and Bastiaanssen, 2013) and Multi-Source Weighted-Ensemble Precipitation (MSWEP) method (Beck et al., 2017) have been widely used.*".

Reference:

Beck, H. E., van Dijk, A. I. J. M., Levizzani, V., Schellekens, J., Miralles, D. G., Martens, B., and de Roo, A.: MSWEP: 3-hourly 0.25° global gridded precipitation (1979–2015) by merging gauge, satellite, and reanalysis data, Hydrol. Earth Syst. Sci., 21, 589–615, https://doi.org/10.5194/hess-21-589-2017, 2017.

Beck, H. E., Wood, E. F., Pan, M., Fisher, C. K., Miralles, D. G., van Dijk, A. I. J. M., McVicar, T. R., Adler, R. F.: MSWEP V2 Global 3-Hourly 0.1° Precipitation: Methodology and Quantitative Assessment, Bull Am. Meteorol. Soc., 100, 473-500, https://doi.org/10.1175/BAMS-D-17-0138.1, 2019.

L69: Diversified methods?

**Response:**

Thank you for pointing this out. We have changed "Diversified data merging methods" to "Various data merging methods" in the revised manuscript. The text there reads as: "*Various data merging methods, such as Kalman filtering algorithm (Pipunic et al., 2008; Liu C et al., 2013), Bayesian Model Average (BMA) and Empirical Orthogonal Function (EOF), can improve regional ET estimation by merging multiple ET products (Yao et al., 2014, 2016; Feng et al., 2016; Zhu et al., 2016).*".

L73: "determine the conditional density" difficult to understand if not familiar with the method, is there an easier way to say this?

**Response:**

Thank you for that comment. The conditional density function determines the performance of the BMA method (Hoeting et al., 1999; Raftery et al., 2005; Duan and

Phillips, 2010). Normal density is valid for surface energy variables (Wu et al., 2012; Miao et al., 2013; Shi and Liang, 2013), while may not apply to water variables because they have a positive probability and their distributions tend to be skewed (Raftery et al., 2005; Yang et al., 2012). Since land ET is a complex variable coupling energy, hydrology and carbon budget, it is difficult to accurately determine the optimal conditional density function in BMA (Yao et al., 2014). We have modified the description in the revised manuscript. The text there reads as: "*Since land ET is a complex variable coupling energy, hydrology and carbon budget, it is difficult to accurately determine the optimal conditional density function in BMA that determines the performance of the method (Yao et al., 2014).*".

Reference:

Hoeting, J. A., Madigan, D., Raftery, A. E., and Volinsky, C. T.: Bayesian model averaging: A tutorial, Stat. Sci., 14, 382–417, 1999.

Raftery, A. E., Gneiting, T., Balabdaoui, F., and Polakowski, M.: Using Bayesian model averaging to calibrate forecast ensembles, Mon. Weather Rev., 133, 1155–1174, https://doi.org/10.1175/MWR2906.1, 2005.

Duan, Q., and Phillips, T. J.: Bayesian estimation of local signal and noise in multimodel simulations of climate change, J. Geophys. Res., 115, D18123, https://doi.org/10.1029/2009JD013654, 2010.

Wu, H., Zhang, X., Liang, S., Yang, H., and Zhou, G.: Estimation of clear-sky land surface longwave radiation from MODIS data products by merging multiple models, J. Geophys. Res., 117, D22107, https://doi.org/10.1029/2012JD017567, 2012.

Miao, C., Duan, Q., Sun, Q., and Li, J.: Evaluation and application of Bayesian multi-model estimation in temperature simulations, Prog. Phys. Geogr., 37, 727–744, https://doi.org/10.1177/0309133313494961, 2013.

Shi, Q., and Liang, S.: Characterizing the surface radiation budget over the Tibetan Plateau with ground-measured, reanalysis, and remote sensing data sets: 1. Methodology, J. Geophys. Res. Atmos., 118, 9642–9657, https://doi.org/10.1002/jgrd.50720, 2013.

Yang, C., Yan, Z., and Shao, Y.: Probabilistic precipitation forecasting based on ensemble output using generalized additive models and Bayesian model averaging, Acta Meteorol. Sin., 26, 1–12, https://doi.org/10.1007/s13351-012-0101-8, 2012.

Yao, Y., Liang, S., Li, X., Hong, Y., Fisher, J., Zhang, N., Chen, J., Cheng, J., Zhao, S., Zhang, X., Jiang, B., Sun, L., Jia, K., Wang, K., Chen, Y., Mu, Q., and Feng, F.: Bayesian multimodel estimation of global terrestrial latent heat flux from eddy covariance, meteorological, and satellite observations, J. Geophys. Res-Atmos., 119, 4521-4545, https://doi.org/10.1002/2013JD020864, 2014.

L74: Is computational effort really a problem for these methods in the age of HPC and cloud computing? Just a thought.

**Response:**

We appreciate your advice. Computational effort is not a problem to some extent.

We have modified the description in the revised manuscript according to the comment. The text there reads as: "*In addition, EOF introduces bias due to the lack of distinction between good and bad quality pixels in the refactoring scheme.*".

L81: I think TC also requires strictly independent datasets and normally distributed errors which is seldom the case? Please double-check.

**Response:**

Thank you for your very thoughtful comment. TC also requires that the errors generated by three datasets must be independent, orthogonal and uncorrelated (McColl et al., 2014; Su et al., 2014a,b; Gruber et al., 2016).

Reference:

Mccoll, K. A., Vogelzang, J., Konings, A. G., Entekhabi, D., Piles, M., Stofffffelen, A.: Extended triple collocation: estimating errors and correlation coeffiffiffficients with respect to an unknown target, Geophys. Res. Lett., 41, 6229-6236, http://dx.doi.org/10.1002/2014GL061322, 2014.

Su, C., Ryu, D., Crow, W. T., Western, A. W.: Remote sensing of environment standalone error characterisation of microwave satellite soil moisture using a fourier method, Remote Sens. Environ., 154, 115-126, http://dx.doi.org/10.1016/j.rse.2014. 08.014, 2014a.

Su, C. H., Ryu, D., Crow, W. T., Western, W. A.: Beyond triple collocation: applications to soil moisture monitoring, J. Geophys. Res.: Atmos., 119, 6419-6439, http://dx.doi.org/10.1002/2013JD021043, 2014b.

Gruber, A., Su, C., Zwieback, S., Crow, W., Dorigo, W., Wagner, W.: Recent advances in (soil moisture) triple collocation analysis, Int. J. Appl. Earth Observ. Geoinf., 45, 200-211, http://dx.doi.org/10.1016/j.jag.2015.09.002, 2016.

L88: Compared to which "simple method"?

**Response:**

Thank you for pointing this out. We have changed "simple method" to "simple average method" in the revised manuscript. The text there reads as: "*Compared with the simple average method*".

L90: Which standards?

**Response:**

Thank you for that comment. We have added the description of the standards in the revised manuscript. The text there reads as: "*These standards including the bias of the simulated ET from the reference and the distance of the simulated ET from the ensemble average are regional rather than global, as most models tend to show anomalous behavior or poor performance from one region to another.*".

L91: "The REA method also produces a quantitative measure of reliability, which increases the overall reliability of simulation changes." Didn't understand this.

**Response:**

Thank you for pointing this out. We have modified the description in the revised manuscript. The text there reads as: "*The REA method also produces a quantitative measure of reliability, indicating that the simulations need to meet both criteria in order to improve the overall reliability of simulation changes.*".

L99: "However, studies on the application of land evaporation are few. Which ones?

**Response:**

Thank you for pointing this out. The "studies" here refers to the study of the application of REA method in terms of land evaporation merging. We have modified the description in the revised manuscript. The text there reads as: "*However, there are few studies on the application of area-averaged grid-scale merging of long sequence model-based land evaporation data.*".

L109: Why is GLEAM the only independent dataset? Actually, the "a" version of GLEAM uses reanalysis data as inputs and therefore will be highly correlated with the others, especially ERA.

**Response:**

Thank you for your cogent advice. GLEAM is a long sequence data set predominantly based on remote sensing observations, and on occasion, reanalysis data. GLEAM is unlike traditional land models, such as found in ERA5, MERRA2 and GLDAS, in that it is driven by satellite observations to obtain evaporation estimates. The version of GLEAM here relies very little on reanalysis datasets (only radiation and temperature of ERA-Interim). Although the "b" version is based entirely on satellite observations, its length of time period is too short to achieve the purpose. Therefore, GLEAM has the most independence relative to the model-based products. GLEAM would be expected to correlate with ERA5 to a certain degree that would warrant thorough

analysis to actually quantify. This is because only the radiation and temperature inputs are from reanalysis products, in this case, ERA Interim, which has been noted in several publications to have some appreciable differences with ERA5 which comes with several updates and changes from its predecessor. Secondly, since GLEAM is predominantly driven by satellite observations, it is expected that the impacts of the reanalysis inputs would not have a direct impact on the output of the merging process. Given these reasons, we believe that ERA5 would reliably be useful in the study, especially over areas significant uncertainties may be found in the reference.

The model based products do employ several satellite inputs through data assimilation to update the prognostic states of the models. MERRA2 and ERA5 are based on brightness temperatures that are assimilated into their atmospheric models, and only indirectly impact the land states. Additionally, the climate models, as well as data assimilations schemes of the two are significantly different. Thus, we can expect some clear differences as a result of the different parameterization schemes. GLDAS, on the other hand, is not a reanalysis product, but uses updated atmospheric forcings from an earlier run which are used to drive the land model to obtain its outputs. Therefore, it does hold some independence from the other two. Eventually, the aim is that the different strengths from these models would be leveraged in the combination. Nonetheless, we do expect some cross-correlation errors to a certain degree.

We have added the description of the relative independence of GLEAM in the revised manuscript. The text there reads as: "*In addition, GLEAM is a long sequence data set predominantly based on remote sensing observations, and on occasion, reanalysis data. GLEAM is unlike traditional land models, such as found in ERA5, MERRA2 and GLDAS, in that it is driven by satellite observations to obtain evaporation estimates. The version of GLEAM here relies very little on reanalysis datasets (only radiation and temperature of ERA-Interim). Therefore, GLEAM has the most independence relative to the model-based products, which is selected as the reference data due to its relative independence.*".

L105 contradicts L111: Is GLEAM merged or not?

**Response:**

Thank you for pointing this out. GLEAM was used as the reference data and was not included in the merging. We have corrected the description in the revised manuscript. The text there reads as: "*Three widely used land ET data sets were selected for merging, including the fifth-generation ECMWF Re-Analysis (ERA5; Hersbach et al., 2020), the second Modern-Era Retrospective analysis for Research and Applications (MERRA2; Gelaro et al., 2017), and Global Land Data Assimilation System ET (GLDAS; Sheffield & Wood, 2007). The differences in spatial and temporal resolution among the ET products were rescaled to a daily timescale and 0.25°, with the time*

*span from 1980 to 2017. Global Land Evaporation Amsterdam Model (GLEAM; Miralles et al., 2011b) was used as the reference data due to its relative independence from other data sets participating in the merging process.*".

"*The data involved in merging includes ERA5, MERRA2 and GLDAS.*".

L116: "the parameterized physical process" Which ones? A few more lines might be helpful here. Which extensive remote sensing observations? Actually, the "a" version uses a lot of model input data. There is an updated v35 version available just for information.

**Response:**

Thank you for that comment. We have added the parameterized physical process in the revised manuscript. The text there reads as: "*GLEAM algorithm estimates land evaporation mainly based on the parameterized physical process. Stress conditions are parameterized as a function of dynamic vegetation information and available water in the root zone. In addition, the detailed parameterization of forest interception is one of its key features. Canopy interception loss, a component of land evaporation, is calculated by the daily precipitation using the parameters describing canopy storage, canopy coverage, and average precipitation and evaporation rate under saturated canopy conditions.*"

GLEAM v3a is based on satellite and reanalysis data. Extensive remote sensing observations includes snow-water equivalent, vegetation optical depth and soil moisture. The version of GLEAM here relies very little on reanalysis datasets (only radiation and temperature of ERA-Interim). Although the "b" version is based entirely on satellite observations, its length of time period is too short to achieve the purpose. We have added the description of these remote sensing observations in the revised manuscript. The text there reads as: "*It uses extensive independent remote sensing observations such as snow-water equivalent, vegetation optical depth and soil moisture as the basis for calculating land evaporation and its different components, including transpiration, bare-soil evaporation, interception loss, open-water evaporation and sublimation separately (Priestley and Taylor, 1972).*".

The updated v35 version is likely to be a better choice. Compared with v3.2a, v3.5a uses the latest version of MSWEP precipitation (v2.8), ESA-CCI soil moisture (v5.3), and VODCA VOD. However, it satisfied the need to be the reference data because GLEAM v3 has been validated with the observations obtained from the eddy covariance instrument on a global scale, which can be used to describe terrestrial ET in different ecosystems (Miralles et al., 2011b). If necessary, we will use v35 for merging and compare with the current results.

L144: Why monthly data? This would have a significant impact but is not really mentioned anywhere else.

**Response:**

Thank you for pointing this out. We have deleted the sentence and added another statement. The sentence was intended to express the quality of MERRA2 data was guaranteed, but it caused confusion. We have added a description of the validation for MERRA2 in the revised manuscript. The text there reads as: "*The accuracy of MERRA2 has been widely evaluated (Bosilovich et al., 2015; Gelaro et al., 2017), including water cycle variability and the global water balance (Bosilovich et al., 2017).*".

Reference:

Bosilovich, M. G., Robertson, F., Takacs, L., Molod, A., and Mocko, D.: Atmospheric water balance and variability in the MERRA-2 reanalysis, J. Climate, 30, 1177–1196, https://doi.org/10.1175/JCLI-D-16-0338.1, 2017.

L155: Why only monthly? How does this affect the method, already mentioned here?

**Response:**

Thank you for that comment. Both daily and monthly data have been used. We have corrected the description in the revised manuscript. The text there reads as: "*Therefore, both daily and monthly land evaporation data of GLDAS2 combined with Noah LSM (GLDAS2-Noah) has been used in this study, whose spatial resolution is 0.25°×0.25°.*".

L179: "based on GLEAM" What is meant by this?

**Response:**

Thank you for pointing this out. "Based on GLEAM" refers to used GLEAM as the reference. We have modified the description in the revised manuscript. The text there reads as: "*Therefore, three data sets have been weighted with GLEAM as the reference, and the performance of the merged ET products have been studied at the selected sites.*".

2.2.1 Should this be "Coefficient of Variation"?

**Response:**

Thank you for your cogent advice. We have changed "Variable Coefficient" to "Coefficient of Variation" in the revised manuscript. The text there reads as: "*2.2.1 Coefficient of Variation*".

L201: So far this raises the biggest question mark for me: Why is GLEAM used as an independent reference dataset. Firstly, it will be highly correlated with the other datasets as it relies on similar inputs, especially the (a) version used here which is forced by ERA-Interim. Although it is a model specifically designed to estimate evaporation, whereas the other models are optimised towards a plethora of variables, it is, in the end, a similar, albeit more simple and specific, model.

**Response:**

Thank you for your very thoughtful comment. GLEAM is a long sequence data set predominantly based on remote sensing observations, and on occasion, reanalysis data. GLEAM is unlike traditional land models, such as found in ERA5, MERRA2 and GLDAS, in that it is driven by satellite observations to obtain evaporation estimates. The version of GLEAM here relies very little on reanalysis datasets (only radiation and temperature of ERA-Interim). Although the "b" version is based entirely on satellite observations, its length of time period is too short to achieve the purpose. Therefore, GLEAM has the most independence relative to the model-based products. GLEAM would be expected to correlate with ERA5 to a certain degree that would warrant thorough analysis to actually quantify. This is because only the radiation and temperature inputs are from reanalysis products, in this case, ERA Interim, which has been noted in several publications to have some appreciable differences with ERA5 which comes with several updates and changes from its predecessor. Secondly, since GLEAM is predominantly driven by satellite observations, it is expected that the impacts of the reanalysis inputs would not have a direct impact on the output of the merging process. Given these reasons, we believe that ERA5 would reliably be useful in the study, especially over areas significant uncertainties may be found in the reference.

The model based products do employ several satellite inputs through data assimilation to update the prognostic states of the models. MERRA2 and ERA5 are based on brightness temperatures that are assimilated into their atmospheric models, and only indirectly impact the land states. Additionally, the climate models, as well as data assimilations schemes of the two are significantly different. Thus, we can expect some clear differences as a result of the different parameterization schemes. GLDAS, on the other hand, is not a reanalysis product, but uses updated atmospheric forcings from an earlier run which are used to drive the land model to obtain its outputs. Therefore, it does hold some independence from the other two. Eventually, the aim is that the different strengths from these models would be leveraged in the combination.

Nonetheless, we do expect some cross-correlation errors to a certain degree.

We have added the description of the relative independence of GLEAM in the revised manuscript. The text there reads as: "*In addition, GLEAM is a long sequence data set predominantly based on remote sensing observations, and on occasion, reanalysis data. GLEAM is unlike traditional land models, such as found in ERA5, MERRA2 and GLDAS, in that it is driven by satellite observations to obtain evaporation estimates. The version of GLEAM here relies very little on reanalysis datasets (only radiation and temperature of ERA-Interim). Therefore, GLEAM has the most independence relative to the model-based products, which is selected as the reference data due to its relative independence.*".

L222: "and ensemble mean" Clarify that ensemble mean refers to the mean of the three products (it's quite a small ensemble).

**Response:**

Thank you for your cogent advice. We have changed "ensemble mean" to "the mean of the three products" in the revised manuscript. The text there reads as: "*The consistency of the three data sets has been illustrated in Fig. 2, where Fig. 2a-c shows the differences of land evaporation between each data set and the mean of the three products.*".

L223: "GLDAS Noah 2 ET is more than 20% higher than the ensemble mean, while ERA5 ET is almost the same with it, even MERRA2 Et more than 20% lower." Please rephrase the sentence.

**Response:**

We appreciate your advice. We have rephrased the sentence in the revised manuscript. The text there reads as: "*In the high latitudes of the northern hemisphere, GLDAS-Noah2 ET is more than 20% higher than the mean of the three products, and ERA5 ET is almost the same as the mean, while MERRA2 ET is more than 20% lower than the mean.*".

L228: "In order to reduce the risk of ...". Which risk?

**Response:**

Thank you for that comment. The risk refers to the accuracy of the estimate. The higher the risk, the lower the accuracy. The large differences among the three products indicate great uncertainty in land evaporation estimation in this region, as a

result that the mean of three products is greatly affected by the accuracy of individual product estimation, which affects the reliability criteria named model convergence of REA method. We have modified the description in the revised manuscript. The text there reads as: "*In order to reduce the risk of inaccuracy in land evaporation merging, CV is used to select regions with high consistency.*".

I'd better justify the choice of excluding some areas in the text. In the end, shouldn't merging especially be important in areas with high discrepancies? As an extreme example: Where the datasets all agree merging is really necessary.

**Response:**

Thank you very much for your comment. These excluded areas are concentrated in hyper-arid areas where some methods for estimating land evaporation are not applicable (Goya and Harmsen, 2014). Many reliable methods, such as physics-based methods, require large amounts of meteorological data, which are not readily available in these areas due to neither adequate coverage nor good quality of these data (Zittis, 2017). If the data we used for merging are highly different from each other and none of them are close to the reference data, merging in these regions does not make sense. For the overall reliability of the merged product, we excluded these areas that might be highly uncertain. It is not uncommon in other studies. For example, the MODIS global evapotranspiration product developed by Mu et al. (2011) excludes barren or sparsely vegetated regions. The soil moisture product developed by Liu et al. (2012) excludes tropical rain forests due to high vegetation density. We have added an explanation for the exclusion of some areas during merging in the revised manuscript. The text there reads as: "*In essence, these excluded areas are concentrated in hyper-arid areas where some methods for estimating land evaporation are not applicable (Goya and Harmsen, 2014). If the data we used for merging are highly different from each other and none of them are close to the reference data, merging in these regions does not make sense. For the overall reliability of the merged product, we excluded these areas that might be highly uncertain.*".

Reference:

Goya, M. R., and Harmsen, E. W.: Evapotranspiration principles and applications for water management, https://doi.org/10.1201/b15779, 2014.

Zittis, G.: Observed rainfall trends and precipitation uncertainty in the vicinity ofthe Mediterranean, Middle East and North Africa Theoretical and Applied, Climatology, 134, 1207-1230, https://doi.org/10.1007/ s00704-017-2333-0, 2017.

Liu, Y. Y., Dorigo, W. A., Parinussa, R. M., de Jeu, R. A. M., Wagner, W., McCabe, M. F., Evans, J. P., van Dijk, A. I. J. M.: Trend-preserving blending of passive and active microwave soil moisture

retrievals, Remote Sensing of Environment, 123, 280-297, https://doi.org/10.1016/j.rse.2012.03.014, 2012.

Are the differences actually very big in absolute terms? Especially in North Africa E is close to zero anyhow.

**Response:**

Thank you for that comment. In absolute terms, the differences are small. Vinukollu et al. (2011) revealed the high uncertainties of land evaporation in the Sahel region, because of the high variability of meteorological variables such as precipitation and radiation, which resulted in large differences among models.

Reference:

Vinukollu, R. K., Meynadier, R., Sheffield, J., and Wood, E. F.: Multi-model, multi-sensor estimates of global evapotranspiration: Climatology, uncertainties and trends, Hydrol. Process, 25, 3993-4010, https://doi.org/10.1002/hyp.8393, 2011.

L273/276: significantly in a statistical sense? Did you compute significance? Could be good to add this.

**Response:**

Thank you for your cogent advice. We have computed significance and added descriptions in the revised manuscript. The text there reads as: "*The correlation coefficients (R) have passed the 5% significance test.*".

L295: Which NDVI product was used for this analysis?

**Response:**

Thank you very much for your comment. Monthly GIMMS NDVI3g data with a spatial resolution of 0.25° was used for the analysis and we have added the description of the product in the revised manuscript. The text there reads as: "*Monthly GIMMS NDVI3g data with a spatial resolution of 0.25° from the Global Inventory Modeling and Mapping Studies (GIMMS) was used in our study (Pinzon & Tucker 2014), with the time span from 1982 to 2014, which is available from http://ecocast.arc.nasa.gov/data/pub/gimms/3g/.*".

Reference:

Pinzon, J. E., and Tucker, C. J.: A non-stationary 1981-2012 AVHRR NDVI3g time series, Remote Sens., 6, 6929-6960, https://doi.org/10.3390/rs6086929, 2014.

L322: The other models include soil moisture too, no? Please double-check.

**Response:**

Thank you very much for your comment. Indeed, GLEAM is not the only one that contains soil moisture, however, GLEAM is the only product that uses satellite retrieved soil moisture to drive the model. The two reanalysis products, ERA5 and MERRA2, depend on atmospheric based observations from satellites and ground observations assimilated into their atmospheric models. GLDAS, on the other hand, is a result of a free model run forced with atmospheric observations from satellites and ground observations. We have deleted the incorrect description and its subsequent paragraph.

L348: Why wasn't the dataset taken into account for this period? This could strongly affect the entire methodology and should be communicated more upfront. Maybe I missed something …

**Response:**

Thank you for that comment. The variation trend of GLDAS-Noah2 has not been taken into account here, which means it has not been calculated for comparison with other data set here, however in the merging process, only the climatology of each data set has been removed, and the variation trends of all data sets are retained. Two versions of GLDAS-Noah2 data are considered in two time periods, GLDAS-Noah2.0 is used from 1980 to 1999, and the newer GLDAS-Noah2.1 is used from 2000 to 2017. Therefore, the variation trend of GLDAS-Noah2 from 1980 to 2017 was not calculated. We have modified the description in the revised manuscript. The text there reads as: "*Figure 9 depicts the variation trends of multiple data sets during 1980-2017, where GLDAS-Noah2 has not been calculated for comparison due to two data sets including GlDAS-NOah2.0 and Gldas-Noah2.1 used throughout the period.*".

We have added the specific steps to detail the merging process in section "2.2.2 Reliability Ensemble Averaging". Anomalies of ERA5, GLDAS and MERRA2 have been merged using GLEAM anomalies as reference. The climatology of ERA5 has been added with the merged anomalies to get the final merged product. The climatology of ERA5 was chosen because its quality was superior to three data sets participating in the merging and the reference data when compared to the EC

measured ET (Fig. 5 & 6). The text in the revised manuscript reads as: "*The specific merging steps are as follows:*

*Step 1: Select the best climatology according to the root mean square deviation (RMSD) between each data set and EC measured ET.*

*Step 2: Calculate the anomalies of each data set participating in the merging and the reference data.*

*Step 3: Merge the anomalies.*

*Step 4: Add the best climatology to the merged anomalies to get the final merged product.*".

L405: 0.5 degrees?

**Response:**

Thank you for pointing this out. We have changed "0.5 degree" to "0.25 degree" in the revised manuscript. The text there reads as: "*We merged three land ET data sets, ERA5, GLDAS and MERRA2, respectively using REA method to generate a set of long sequence global daily ET data with a spatial resolution of 0.25 degree and a time span of 38 years.*".

---

## Author Comment (AC2)

**Answer to Reviewer 2 ESSD-2021-61**

We thank Reviewer 2 for the comments. We provide here our responses to those comments and describe how we addressed them in the revised manuscript. The original reviewer comments are in normal black font while our answers appear in blue font. The corresponding edit in the manuscript are included in red font.

Lu et al., 2021 present a new merged global land evaporation dataset based on three existing products. While the paper is interesting for publication is this journal, my current recommendation is for a major revision for the following reasons. First, there is the issue of considering GLEAM as an observational dataset. GLEAM is really just another data source, and the authors even show that the individual models they are comparing outperform GLEAM in terms of R and RMSD compared to flux sites. I think the comparisons with GLEAM are okay, as long as authors specifically mention that it is not used for validation.

**Response:**

Thank you for your very thoughtful comment and suggestions. We have added the clarification of the role of GLEAM as an independent reference data set in the data section. GLEAM is a long sequence data set predominantly based on remote sensing observations, and on occasion, reanalysis data. GLEAM is unlike traditional land models, such as found in ERA5, MERRA2 and GLDAS, in that it is driven by satellite observations to obtain evaporation estimates. The version of GLEAM here relies very little on reanalysis datasets (only radiation and temperature of ERA-Interim). Therefore, GLEAM has the most independence relative to the model-based products.

We have added the description of the relative independence of GLEAM in the revised manuscript. The text there reads as: "In addition, GLEAM is a long sequence data set predominantly based on remote sensing observations, and on occasion, reanalysis data. GLEAM is unlike traditional land models, such as found in ERA5, MERRA2 and GLDAS, in that it is driven by satellite observations to obtain evaporation estimates. The version of GLEAM here relies very little on reanalysis datasets (only radiation and temperature of ERA-Interim). Therefore, GLEAM has the most independence relative to the model-based products, which is selected as the reference data due to its relative independence.".

GLEAM was included in preliminary evaluations of the ET estimates. By including GLEAM in the evaluations, we aim to assess regions where the reference data will be potentially less reliable. This also provides the information on regions of high uncertainties with respect to the reference data, which becomes very useful with applications. Thus, we obtained a good understanding of the skill of GLEAM prior to

its use as the reference data. In addition, an aim of the study is to leverage the uniqueness of GLEAM (as discussed above) to combine the model-based products. It is expected that GLEAM's over-reliance on observations states would serve as some sort of benchmark to estimate the weights of the model-based products. Thus, the goal is not based on a superior skill of GLEAM but its added value due to its uniqueness relative to the model-based products, which we believe, does have merits.

My other major concerns are some inconsistencies I see in figures, that need to be reevaluated or at least better explained (see more detailed comments below). One example in Figure 7, high NDVI values are incorrectly linked solely to humidity conditions, and the authors do not discuss the potential saturation issues at high NDVI values often seen in remote sensing datasets. The authors omit some necessary details (i.e. the use and source of NDVI is never explained in methods). Lastly, there are some grammar errors which should be addressed.

**Response:**

Thank you very much for your comment. We have made more specific explanations of the corresponding problems. Thereinto, we have corrected the one-sided expression about high NDVI values in the revised manuscript. The text there reads as: "As shown in Fig. 7, the quality of each data set is relatively low and shows a rapid decline with the increase of vegetation density when NDVI is greater than 0.7, the case of optimal conditions for vegetation growth.".

We have added the comments on the potential issue of vegetation index saturation at high amounts of NDVI to the manuscript. Vegetation index saturation at high amounts of NDVI poses potential issues. Generally speaking, NDVI is likely to become saturated over a dense canopy for forested areas, and becomes saturated rapidly for vegetation with a nearly closed canopy (Liu et al., 2011). Based on the analysis of hyperspectral data, it is found that there is an obvious saturation problem in the relationship between LAI and NDVI, that is, when LAI exceeds 2, NDVI asymptotically reaches the saturation level (Haboudane et al., 2004). When biomass reaches a certain level, NDVI is not sensitive to changes in biomass (Huang et al., 2021). Dynamic vegetation is not used in these models, resulting in lower data quality with dense vegetation. Therefore, vegetation index saturation at high amounts of NDVI results in a decrease in the quality of these datasets at high vegetation density. As shown in Fig. 7, the quality of each data set is relatively low and shows a rapid decline with the increase of vegetation density when NDVI is greater than 0.7, the case of optimal conditions for vegetation growth. In addition, a lot of remote sensing data have been used in GLEAM, such as satellite soil moisture, which is not of high quality when the vegetation density is high, affecting the quality of the final

output. Further, errors in GLEAM will affect the merged product because GLEAM acts as the reference data. The text in the revised manuscript reads as: "It is well known that vegetation index saturation poses potential issues. Generally speaking, NDVI is likely to become saturated over a dense canopy for forested areas, and becomes saturated rapidly for vegetation with a nearly closed canopy (Liu et al., 2011). Based on the analysis of hyperspectral data, it is found that there is an obvious saturation problem in the relationship between LAI and NDVI, that is, when LAI exceeds 2, NDVI asymptotically reaches the saturation level (Haboudane et al., 2004). When biomass reaches a certain level, NDVI is not sensitive to changes in biomass (Huang et al., 2021). Dynamic vegetation is not used in these models, resulting in lower data quality with dense vegetation. Therefore, vegetation index saturation at high amounts of NDVI results in a decrease in the quality of these datasets at high vegetation density. As shown in Fig. 7, the quality of each data set is relatively low and shows a rapid decline with the increase of vegetation density when NDVI is greater than 0.7, the case of optimal conditions for vegetation growth. In addition, a lot of remote sensing data have been used in GLEAM, such as satellite soil moisture, which is not of high quality when the vegetation density is high, affecting the quality of the final output. Further, errors in GLEAM will affect the merged product because GLEAM acts as the reference data.".

Monthly GIMMS NDVI3g data with a spatial resolution of 0.25° was used for the analysis and we have added the description of the product in the revised manuscript. The text there reads as: "Monthly GIMMS NDVI3g data with a spatial resolution of 0.25° from the Global Inventory Modeling and Mapping Studies (GIMMS) was used in our study (Pinzon & Tucker 2014), with the time span from 1982 to 2014, which is available from http://ecocast.arc.nasa.gov/data/pub/gimms/3g/.".

We have corrected the grammar errors.

**Reference:**

- Liu, Y. Y., de Jeu, R. A. M., McCabe, M. F., Evans, J. P., and van Dijk, A. I. J. M.: Global long-term passive microwave satellite-based retrievals of vegetation optical depth, Geophys. Res. Lett., 38, L18402, https://doi.org/10.1029/2011GL048684, 2011.
- Haboudanea, D., Miller, J. R., Pattey, E., Zarco-Tejada, P. J., and Strachan, I. B.: Hyperspectral vegetation indices and novel algorithms for predicting green LAI of crop canopies: Modeling and validation in the context of precision agriculture, Remote Sens. Environ., 90, 337–352, https://doi.org/10.1016/j.rse.2003.12.013, 2004.
- Huang, S., Tang, L., Hupy, J. P., Wang, Y., and Shao, G.: A commentary review on the use of normalized difference vegetation index (NDVI) in the era of popular remote sensing, J. For. Res., 32, 1–6, https://doi.org/10.1007/s11676-020-01155-1, 2021.

Line 36: Since 'the' land surface

**Response:**

Thank you for pointing this out. We have corrected it.

Line 37: resulted should be resulting

**Response:**

Thank you for pointing this out. We have corrected it.

Line 46: Should say flux tower data

**Response:**

Thank you for pointing this out. We have corrected it.

Line 74: Pixel should not be capitalized

**Response:**

Thank you for pointing this out. We have corrected it.

Line 82: Lately for Lastly

**Response:**

Thank you for pointing this out. We have corrected it.

Line 88-90: What is the reference for this?

**Response:**

Thank you for that comment. We have added the reference in the revised manuscript. The text there reads as: "Compared with the simple average method, Reliability Ensemble Average method (REA) extracts the most reliable information from each model by minimizing the impact of "outliers" or underperforming models, subsequently reducing the uncertainty range in simulated changes, which also stands out in terms of computational efficiency (Giorgi & Mearns, 2002)." Table 1: If using GLEAMv3, should include the Martens et al., 2017 reference (https://doi.org/10.5194/gmd-10-1903-2017)

**Response:**

Thank you very much for your comment. We have added the reference in the revised manuscript. The text there reads as: "

| Table 1. Summary of ET data sets involved in merging. |                                        |                                   |                        |                         |                                                        |
|-------------------------------------------------------|----------------------------------------|-----------------------------------|------------------------|-------------------------|--------------------------------------------------------|
| Name                                                  | ET schemes/
land-surface
schemes | Spatial
resolution
(degree) | Temporal
resolution | Time span               | Reference                                              |
| GLEAM3.2a                                             | Priestley-Taylor                       | 0.25×0.25                         | daily                  | 1980-2017               | Miralles et al.
(2011b)
Martens et al.
(2017) |
| ERA5                                                  | IFS                                    | 0.25×0.25                         | 1-hour                 | 1980-2017               | Hersbach et al.
(2020)                              |
| MERRA2                                                | GEOS-5                                 | 0.625×0.5                         | daily                  | 1980-2017               | Gelaro et al.
(2017)                                |
| GLDAS2.0 &
2.1                                     | Noah                                   | 0.25×0.25                         | 3-hour                 | 1980-1999&
2000-2017 | Sheffield &
Wood (2007)                             |

".

Reference:

Martens, B., Miralles, D. G., Lievens, H., van der Schalie, R., de Jeu, R. A. M., Fernández-Prieto, Diego., Beck, H. E., Dorigo, W. A., Verhoest, N. E. C.: GLEAM v3: satellite-based land evaporation and root-zone soil moisture, Geosci. Model Dev., 10, http://doi.org/1903–1925, 2017.

Line 119: Can you give some examples of the empirical parameters you mean? If not, might be best to remove.

**Response:**

Thank you for your cogent advice. There are some empirical parameters such as the evaporation stress factor (S), the latent heat of evaporation ( $\lambda$ ) and the slope of the saturated water vapour-temperature curve ( $\Delta$ ).

Phenological constraints, heat stress or water availability affecting evaporation are usually combined in a empirical stress factor (Sellers et al., 2007). In GLEAM, an empirical parameter called evaporation stress factor (S) is defined, which ranges between 0 (maximum stress and no evaporation) and 1(no stress and potential evaporation).

In the Priestley and Taylor (1972) equation, the latent heat of evaporation ( $\lambda$ ) and the slope of the saturated water vapour-temperature curve ( $\Delta$ ) are estimated from an empirical relationship with temperature (Henderson-Sellers, 1984; Maidment, 1993).

$$\lambda E_{\rm p} = \alpha \frac{\Delta}{\Delta + \psi} (R_{\rm n} - G)$$

We have added several specific empirical parameters in the revised manuscript. The text there reads as: "The empirical parameters contained in this algorithm such as the evaporation stress factor, the latent heat of evaporation and the slope of the saturated water vapour-temperature curve have been obtained from the findings in different fields.".

Section 2.1.2. Perhaps it's worth to mention that ERA-5 still appears to overestimate the latent heat flux, https://gmd.copernicus.org/articles/13/4159/2020/

**Response:**

Thank you very much for your comment. We have added it in the revised manuscript. The text there reads as: "Martens et al. (2020) evaluated surface energy partitioning in ERA5 especially including the latent heat fluxes using different reference datasets and modeling tools, with the analysis showing that there is lower absolute biases in the surface latent heat flux of ERA5 than ERA-Interim, though ERA5 still appears to overestimate the latent heat flux in most catchments.".

Line 168: three is spelt out and then indicated by numeric.

**Response:**

Thank you for pointing this out. We have corrected it.

Section 2.1.5: was any masking done to the dataset? I.e. removal of snowy, frozen or rainy days?

**Response:**

Thank you very much for your comment. We masked measurements using the provided quality flags. The removal of snowy, frozen and rainy days was not applied apart from the quality control applied. Generally, eddy-covariance measurements are less reliable during these days. As can be seen from the validation results, monthly scale validation results of all the datasets were better than the daily scale, indicating the reduced reliability has a greater impact on daily scale datasets. Nevertheless, the quality of the merged product is higher than other datasets at the daily scale. Therefore, the validation results are representative. We have added the description of the process of EC ET data preprocessing to section 2.1.5 in the revised manuscript. The text there reads as: "Measurements are masked with the provided quality flags in the data set archives.".

Figure 2&3: While I can understand Figure 2, I am not sure how to interpret it relative to Figure 3. It appears in Figure 2, that over some regions (i.e. the Amazon), the three datasets are in good agreement (CV close to 0). I would think this translates to evenly distributing the weight of each dataset on the merged product, but Figure 3 shows that MERRA2 is much less considered. Some patterns do make sense, for example in northern latitudes, it appears ERA5 is most closely related to the other datasets, despite their being greater CV, so it's more weighted.

**Response:**

Thank you for your very thoughtful comment. The CV analysis aims to evaluate the relative systematic differences between the three model products. As a consequence, relative deviations can be obtained. Since the CV is not computed relative to the reference data, GLEAM, it does not directly translate into the merging approach. This is why these differences between Figure 2a-c & Figure 2d mainly exist. While Figure 2a-c describe the products' contributions due to it a computed weight relative to the GLEAM, CV aims to understand the relative systematic differences apart from the reference. Nonetheless, what we attempt to achieve with the CV analysis is to identify the regions of significant differences between the products even apart from the reference. This serves as some sort of dual check for higher consistencies in the merging scheme.

We have added this explanation to the discussion in the revised manuscript. The text there reads as: "The CV analysis aims to evaluate the relative systematic differences between the three model products. Since it does not take the reference data into account, it does not directly translate into the merging scheme. Nonetheless, it serves as an added check to obtain optimum consistencies in the merging process for higher skill in the merged data." Figure 4:How can the authors explain the nearly symmetrical -50 to 50 mm/year the GLEAM model shows, versus the anomalies in the other products never going below 0?

**Response:**

Thank you for your cogent advice. The bandwidth of the kernel smoothing window was set to 10 to smooth the curve, making the GLEAM model show the nearly symmetrical -50 to 50 mm/year. The anomalies of all the five datasets were obtained by subtracting the climatology of GLEAM ET rather than their own from the original data to highlight the differences between them as a whole. However, we have found that it made more sense to subtract their own climatology than to subtract GLEAM's. Therefore, we have modified Figure 4c and added the bandwidth of the kernel smoothing window to the caption of figure in the revised manuscript.

90° N ERA5 60° N GLDAS2-Noah GLEAM3.2a MERRA2 30° N REA EQ 30° S (a) 60° S 0 300 600 900 1200 1500 Mean land evaporation (mm year-1) **Evapotranspiration Anomalies** 40 ERA5 GLEAM3.2a REA GLDAS2-Noah MERRA2 20 (mm year-1) 0 -20 (b) 1980 1984 1988 1992 1996 2000 2012 2004 2008 2016 4 ERA5 3 GLDAS2-Noah DF (%) GLEAM3.2a 2 MERRA2 REA 1 (c) 0 -40 -20 0 -60 20 40 60 Land Evaporation Anomalies (mm year-1)

The text there reads as: "

Figure 4. a) Latitudinal distribution of mean land evaporation from five data sets, b) time series (1980-2017), and c) probability distribution of annual land evaporation anomalies from five ET products. The bandwidth of the kernel smoothing window was

**set to 10.".**

We have modified the analysis of Figure 4c in the revised manuscript. The text there reads as: "The obvious differences between the probability density distributions of multiple data sets are clearly visible. In general, the consistency between the merged product and GLEAM ET is relatively better, which may be greatly related to GLEAM as the reference data in the merging process. Due to the discrepancies in the driving data and calculation formulations for land evaporation, anomalies vary from data to data.".

Figure 5: This figure highlights one point which is that GLEAM does not even perform better as some of the individual models for tower comparisons. If it is only used for comparisons, and not used as a validation source, that is still acceptable.

**Response:**

Thank you for that comment. GLEAM was included in preliminary evaluations of the ET estimates. By including GLEAM in the evaluations, we aim to assess regions where the reference data will be potentially less reliable. This also provides the information on regions of high uncertainties with respect to the reference data, which becomes very useful with applications. Thus, we obtained a good understanding of the skill of GLEAM prior to its use as the reference data.

Figure 7: The authors do not state where NDVI was obtained from and at what resolution.

**Response:**

Thank you very much for your comment. Monthly GIMMS NDVI3g data with a spatial resolution of 0.25° was used for the analysis and we have added the description of the product in the revised manuscript. The text there reads as: "Monthly GIMMS NDVI3g data with a spatial resolution of 0.25° from the Global Inventory Modeling and Mapping Studies (GIMMS) was used in our study (Pinzon & Tucker 2014), with the time span from 1982 to 2014, which is available from http://ecocast.arc.nasa.gov/data/pub/gimms/3g/.".

Line 303: NDVI >0.7 is not only under humid conditions, rather just optimal conditions for vegetation growth which widely vary depending on the ecosystem.

**Response:**

Thank you for your cogent advice. We have corrected the one-sided expression in the revised manuscript. The text there reads as: "As shown in Fig. 7, the quality of each data set is relatively low and shows a rapid decline with the increase of vegetation density when NDVI is greater than 0.7, the case of optimal conditions for vegetation growth.".

Figure 7: Can the authors comment on the potential issue of vegetation index saturation at high amounts of NDVI or LAI? Could that also explain some of these patterns?

**Response:**

Thank you for your very thoughtful comment. We have added the comments on the potential issue of vegetation index saturation at high amounts of NDVI to the manuscript. Vegetation index saturation at high amounts of NDVI poses potential issues. Generally speaking, NDVI is likely to become saturated over a dense canopy for forested areas, and becomes saturated rapidly for vegetation with a nearly closed canopy (Liu et al., 2011). Based on the analysis of hyperspectral data, it is found that there is an obvious saturation problem in the relationship between LAI and NDVI, that is, when LAI exceeds 2, NDVI asymptotically reaches the saturation level (Haboudane et al., 2004). When biomass reaches a certain level, NDVI is not sensitive to changes in biomass (Huang et al., 2021). Dynamic vegetation is not used in these models, resulting in lower data quality with dense vegetation. Therefore, vegetation index saturation at high amounts of NDVI results in a decrease in the quality of these datasets at high vegetation density. As shown in Fig. 7, the quality of each data set is relatively low and shows a rapid decline with the increase of vegetation density when NDVI is greater than 0.7, the case of optimal conditions for vegetation growth. In addition, a lot of remote sensing data have been used in GLEAM, such as satellite soil moisture, which is not of high quality when the vegetation density is high, affecting the quality of the final output. Further, errors in GLEAM will affect the merged product because GLEAM acts as the reference data. The text in the revised manuscript reads as: "It is well known that vegetation index saturation poses potential issues. Generally speaking, NDVI is likely to become saturated over a dense canopy for forested areas, and becomes saturated rapidly for vegetation with a nearly closed canopy (Liu et al., 2011). Based on the analysis of hyperspectral data, it is found that there is an obvious saturation problem in the relationship between LAI and NDVI, that is, when LAI exceeds 2, NDVI asymptotically reaches the saturation level (Haboudane et al., 2004). When biomass reaches a certain level, NDVI is not sensitive to changes in biomass (Huang et al., 2021). Dynamic vegetation is not used in these models, resulting in lower data quality

with dense vegetation. Therefore, vegetation index saturation at high amounts of NDVI results in a decrease in the quality of these datasets at high vegetation density. As shown in Fig. 7, the quality of each data set is relatively low and shows a rapid decline with the increase of vegetation density when NDVI is greater than 0.7, the case of optimal conditions for vegetation growth. In addition, a lot of remote sensing data have been used in GLEAM, such as satellite soil moisture, which is not of high quality when the vegetation density is high, affecting the quality of the final output. Further, errors in GLEAM will affect the merged product because GLEAM acts as the reference data."

**Reference:**

- Liu, Y. Y., de Jeu, R. A. M., McCabe, M. F., Evans, J. P., and van Dijk, A. I. J. M.: Global long-term passive microwave satellite-based retrievals of vegetation optical depth, Geophys. Res. Lett., 38, L18402, https://doi.org/10.1029/2011GL048684, 2011.
- Haboudanea, D., Miller, J. R., Pattey, E., Zarco-Tejada, P. J., and Strachan, I. B.: Hyperspectral vegetation indices and novel algorithms for predicting green LAI of crop canopies: Modeling and validation in the context of precision agriculture, Remote Sens. Environ., 90, 337–352, https://doi.org/10.1016/j.rse.2003.12.013, 2004.
- Huang, S., Tang, L., Hupy, J. P., Wang, Y., and Shao, G.: A commentary review on the use of normalized difference vegetation index (NDVI) in the era of popular remote sensing, J. For. Res., 32, 1–6, https://doi.org/10.1007/s11676-020-01155-1, 2021.

Line 306: This is true, also it brings the question of what land cover classifications are assigned for each model. If some models for example are using MODIS IGBP versus another data source, this could be a huge reason for discrepancies.

**Response:**

Thank you very much for your comment. We have added this consideration in the revised manuscript. The text there reads as: "There are unique advantages and limitations of the existing land ET data sets for specific land cover types, however, quite few are globally suitable for meteorology and hydrology. Specific land cover classifications are assigned for each model, leading to the use of land cover classification from different sources bringing about discrepancies in the estimation of land ET.".

Line 322: GLEAM is not the only product from even this study which considers soil moisture estimates from satellites.

**Response:**

Thank you for that comment. Indeed, GLEAM is not the only one that contains soil moisture, however, GLEAM is the only product that uses satellite retrieved soil moisture to drive the model. The two reanalysis products, ERA5 and MERRA2, depend on atmospheric based observations from satellites and ground observations assimilated into their atmospheric models. GLDAS, on the other hand, is a result of a free model run forced with atmospheric observations from satellites and ground observations. We have deleted the incorrect description and its subsequent paragraph.

Figure 8: Why are there missing areas in REA which is not observed in the other datasets? Especially in Northern Africa and Asia?

**Response:**

Thank you very much for your comment. We used the coefficient of variation (CV) as the indicator to select the merging regions with high data consistency, and the regions with low consistency were excluded from the merging scope, including the north of North America, west of South America, desert regions of mid-latitude Africa and Asia. The CV analysis aims to evaluate the relative systematic differences between the three model products. As a consequence, relative deviations can be obtained. Nonetheless, what we attempt to achieve with the CV analysis is to identify the regions of significant differences between the products even apart from the reference. This serves as some sort of dual check for higher consistencies in the merging scheme.

We have added this explanation to the discussion in the revised manuscript. The text there reads as: "The CV analysis aims to evaluate the relative systematic differences between the three model products. Since it does not take the reference data into account, it does not directly translate into the merging scheme. Nonetheless, it serves as an added check to obtain optimum consistencies in the merging process for higher skill in the merged data."

Line 405: 0.5 degree or 0.25 degree?

**Response:**

Thank you for pointing this out. We have changed "0.5 degree" to "0.25 degree" in the revised manuscript. The text there reads as: "We merged three land ET data sets, ERA5, GLDAS and MERRA2, respectively using REA method to generate a set of long sequence global daily ET data with a spatial resolution of 0.25 degree and a time span of 38 years.".

---

## Author Response (AR2)

**Editor's comments:**

Referee #1 still has some concerns about the use and exact description of GLEAM, specifically what it is and is not, which I share. I therefore kindly ask you to revise the manuscript once again, carefully considering all remaining comments of the referee.

Thank you for your very thoughtful summary and refinement of the reviewer' comments. We have revised the manuscript carefully according to Referee #1's comments.

**Answer to Reviewer 1 ESSD-2021-61**

We thank Reviewer 1 for the comments. We provide here our responses to those comments and describe how we addressed them in the revised manuscript. The original reviewer comments are in normal black font while our answers appear in blue font. The corresponding edit in the manuscript are included in red font.

Table 2 and 3: I'd strongly suggest to include GLEAM validation metrics.
Why is no validation of GLEAM included? Surely this is vital as GLEAM is very often used for global studies and the paper should provide insights in how, when or where the merged product surpasses GLEAM. Note that when validating E products it is often standard-practise to exclude days with strong precipitation.
If GLEAM outperforms the others for a given pixel, can the merged product actually achieve the same or better performance than the GLEAM reference data? I'm just wondering, some thoughts and a sentence or two would be helpful.

**Response:**

Thank you for your very thoughtful comment and suggestions. We have added the validation metrics of GLEAM in table 2 and 3. The descriptions of table 2 and 3 have been modified in the revised manuscript accordingly. Lines 345-377 read: "

[revised manuscript text omitted]

*Similar to Table 2, Table 3 shows the performance of ET products in different ecosystems on a monthly scale. Compared with daily scale, the performance of each product has changed, among which all of the R becomes higher. REA has higher R and lower RMSD than individual products at 23 DBF, 42 ENF, 8 SAV and 19 WET sites. It has an optimal R or RMSD at 17 CRO, 9 MF and 9 OSH sites. At other 64 sites, REA has a worse performance than at least one individual product. For 1 DNF site, REA has lower R and higher RMSD than ERA5. For 13 EBF sites, REA has lower R and higher RMSD than GLEAM and ERA5. For 34 GRA sites, ERA5 has a lower R of 0.77 than MERRA2 and GLEAM, and a higher RMSD of 27.53 mm per month than GLEAM, ERA5 and GLDAS. For 6 WSA sites, REA has a lower R of 0.73 than GLEAM, and a higher RMSD of 34.49 mm per day than GLEAM, GLDAS and ERA5. Similar to the daily scale, REA does not have a better performance than any individual product in all ecosystems, but is superior to at least one individual one. Similarly, the Taylor charts of monthly Ground-measured ET and ET from the different products in 11 ecosystems are put in support information (Fig. S3).".*

Generally, eddy-covariance measurements are less reliable during days with strong precipitation. We have validated the merged product with Ground-measured ET exclude these days. Fig. S1 shows the taylor diagrams of daily ET from different products and Ground-measured ET including and excluding days with strong precipitation. The table shows the validation metrics including R and RMSD between daily Ground-measured ET and daily ET from different products. As can be seen from the figure and table, there is small difference between the validation results of (a) and (b), which indicating that data on days with strong precipitation have minimal impact on this study. Also, we believe that using the quality control measures provided by the insitu data developers gets us close enough to a representative quality indication. Nonetheless, we have added the some text based on the discussed comparison above to the revised manuscript. Lines 193-195 read: "*Since there is minimal impact of Ground-measured ET on days with strong precipitation on the verification results (Fig. S1), data on these days are not excluded in order to retain more ground data samples for statistical analysis.*".

[Figure]

Figure S1. Taylor diagram of daily Ground-measured ET (a) including, (b) excluding days with strong precipitation and ET from different products.

**Table. The verification results including R and RMSD between daily Ground-measured ET and daily ET from different products.**

|  | ERA5 | | GLDAS | | GLEAM | | MERRA2 | | REA | |
|---|---|---|---|---|---|---|---|---|---|---|
|  | R | RMSD | R | RMSD | R | RMSD | R | RMSD | R | RMSD |
| (a) | 0.69 | 0.96 | 0.66 | 1.12 | 0.66 | 1.03 | 0.66 | 1.18 | **0.72** | **0.91** |
| (b) | 0.68 | 0.97 | 0.65 | 1.10 | 0.66 | 0.99 | 0.65 | 1.15 | **0.70** | **0.92** |

Fig. S6 shows the relationship between the quality of the merged product and GLEAM. As can be seen, the merged product is highly sensitive to the quality of the reference data. If the quality of GLEAM is high, the quality of the merged product will be correspondingly high. We have added the description to the revised manuscript. Lines 519-522 read: "*Consequently, the results also demonstrate that REA is more sensitive to higher qualities in GLEAM. As a result, where GLEAM has lower qualities, REA tends to have higher qualities. Meanwhile, they both have very similar qualities, or even higher in REA, at regions where GLEAM has higher correlations and lower differences with the insitu datasets (Fig S6). Thus REA is more sensitive to the reference data where it is more reliable.*".

[Figure]

Figure S6. Scatter plots of the correlation coefficients and RMSD between GLEAM, REA and Ground-measured ET. Linear fits are plotted in blue and the 1:1 line is depicted.

Furthermore I'd expect a few sentences on the exact input data GLEAM version a relies on based on the respective GLEAM paper. Net radiation as correctly stated is based on ERA-Interim and GLEAM is actually extremely sensitive to net radiation.

**Response:**

Thank you for your cogent advice. GLEAM version 3a is produced using satellite observations including soil moisture, vegetation optical depth and snow-water equivalent, a multi-source precipitation product and relies on only radiation and temperature inputs from reanalysis products (Martens et al., 2017). We have added the description to the revised manuscript. Lines 117-120 read: "*Additionally, GLEAM is not a complex terrestrial model as found in land models of ERA5, MERRA2 and GLDAS, but a set of algorithms dedicated to estimating terrestrial evaporation using retrieved satellite observations including soil moisture, vegetation optical depth and snow-water equivalent, a multi-source precipitation product and relies on only radiation and temperature inputs from reanalysis products (Martens et al., 2017).*".

Concerning these sentences in the rebuttal letter/or text:
Please reword/revise parts in the updated manuscript that reflect on the ideas below.
"GLEAM is not a traditional terrestrial model ..." Please reword this, it's not clear what a 'traditional' terrestrial model is or why GLEAM should be any different. Perhaps argue that GLEAM is specifically designed to estimate evaporation whereas the other 'big' models are required to simulate a higher number of variables decently (This is a spontaneous idea, please check carefully with the literature). GLEAM is not part of a larger Earth System model with an atmospheric/sea ice component etc. Perhaps that's more of a difference too?

**Response:**

Thank you for your very thoughtful comment. We have changed "traditional" to "complex", and added the description of "complex" to the revised manuscript. GLEAM is not a complex terrestrial model as found in land models of ERA5, MERRA2 and GLDAS. It is a set of algorithms dedicated to the estimation of terrestrial evaporation (Martens et al., 2017). Lines 117-120 read: "*Additionally, GLEAM is not a complex terrestrial model as found in land models of ERA5, MERRA2 and GLDAS, but a set of algorithms dedicated to estimating terrestrial evaporation using retrieved satellite observations including soil moisture, vegetation optical depth and snow-water equivalent, a multi-source precipitation product and*

*relies on only radiation and temperature inputs from reanalysis products (Martens et al., 2017).".*

"MERRA2 and ERA5 are based on brightness temperatures that are assimilated into their atmospheric models and only indirectly impact the land states." --> A lot more than brightness temperatures are assimilated, e.g. IR radiances, air temperature measurements from aircraft etc. etc. etc.

**Response:**

Thank you for pointing this out. We have modified the description. This sentence is changed to "MERRA2 and ERA5 are based on brightness temperatures, infrared radiances, air temperature measurements from aircraft and a lot more data that are assimilated into their atmospheric models and only indirectly impact the land states.".

"It is expected that GLEAM's over-reliance on observations states would serve as some sort of benchmark to estimate the weights of the model-based products. Thus, the goal is not based on a superior skill of GLEAM but its added value due to its uniqueness relative to the model-based products, which we believe, does have merits" One might argue that GLEAM is more directly linked to satellite input but it is no less a model than the other products. The reanalysis products incorporate many many more observations than GLEAM does. It is a rather simple model (in a good way) focusing on soil moisture and evaporation (and computes some more variables required for E and soil moisture).

**Response:**

Thank you very much for your comment. It definitely makes a lot of sense this looking at it this way. We have modified the description to "Although GLEAM is no less a model than the other products, the ET output from GLEAM is more directly linked to the satellite retrieval inputs within a more simplified model. This peculiar framework of GLEAM could be reliable to serve as some sort of benchmark from which we estimate the weights of the model-based products. Thus, the goal is not based on a superior skill of GLEAM but its added value due to its uniqueness relative to the other model-based products, which we believe does have merits."

"GLEAM is not a traditional terrestrial model as found in ERA5, MERRA2 and GLDAS" See above, not sure about traditional.

**Response:**

Thank you for your very thoughtful comment. We have changed "traditional" to "complex", and added the description of "complex" to the revised manuscript.

GLEAM is not a complex terrestrial model as found in land models of ERA5, MERRA2 and GLDAS. It is a set of algorithms dedicated to the estimation of terrestrial evaporation (Martens et al., 2017). Lines 117-120 read: "*Additionally, GLEAM is not a complex terrestrial model as found in land models of ERA5, MERRA2 and GLDAS, but a set of algorithms dedicated to estimating terrestrial evaporation using retrieved satellite observations including soil moisture, vegetation optical depth and snow-water equivalent, a multi-source precipitation product and relies on only radiation and temperature inputs from reanalysis products (Martens et al., 2017).*".

"GLEAM (Miralles et al., 2011a) (Global Land-Surface Evaporation: The Amsterdam Methodology) is derived from the inversion of multi-source remote sensing data, meteorological reanalysis data and the improved Priestley-Taylor (P-T) formula"
Surprisingly it is sometimes stated that GLEAM is an inversion or retrieval method but in my view it isn't. Inversion in my understanding is based on forward simulations of something observable from satellite, e.g. brightness temperatures, radiances etc. These forward simulations are based on a model (radiative-transfer) with multiple geophysical input variables. Minimising the difference between forward simulations and a satellite observation by assuming certain geophysical conditions is a retrieval based on inversion. GLEAM does no such thing.
GLEAM is a simple land surface model focusing on the estimation of evaporation and soil moisture. It's a traditional top-down approach with a model being fed with atmospheric input and land surface conditions (e.g. vegetation phenology). The estimation of evaporation is based around the Priestley-Taylor formula.
I'm not sure about the other two evaporation products, I would assume they are also specific models and not inversion schemes at all but please check.

**Response:**

Thank you for your cogent advice. GLEAM and the other two evaporation products are not inversion schemes. We have deleted the following description.

""

"Thank you very much for your comment. Indeed, GLEAM is not the only one that contains soil moisture, however, GLEAM is the only product that uses satellite retrieved soil moisture to drive the model."
Satellite retrieved soil moisture does not drive GLEAM. GLEAM computes soil moisture at different levels based on soil properties, precipitation input etc. very similarly to the other models (similar in principle, not the exact formulas). Satellite

retrievals are assimilated with a very simple Newtonian Nudging scheme slightly correcting the modelled soil moisture. The impact of this assimilation is however mostly quite low. Therefore you can give equal credit to the other models with their repsective soil water modules.

**Response:**

Thank you very much for your comment. We will surely give equal credit to the other models with their respective soil water modules. Our own preliminary studies of their soil moisture modules have shown their commendable skill.

Further comments:
L43: I suppose SiF can be used for E although data quality is still not great (I'm no expert on this).

**Response:**

Thank you for your very thoughtful comment. Recently, solar-induced chlorophyll fluorescence (SIF) has been discovered as an emerging technique to observe the photosynthetic processes of vegetation by quantifying the emission of fluorescent radiation (Joiner et al., 2014). Remotely sensed SIF has potential to empirically track the variation of canopy-level transpiration (Lu et al., 2018; Shan et al., 2019). We have added the description to the revised manuscript. Lines 42-45 read: "*Recently, solar-induced chlorophyll fluorescence (SIF) has been discovered as an emerging technique to observe the photosynthetic processes of vegetation by quantifying the emission of fluorescent radiation (Joiner et al., 2014). Remotely sensed SIF has potential to empirically track the variation of canopy-level transpiration (Lu et al., 2018; Shan et al., 2019).*".

I think it's still missing a clearer justification of using GLEAM (and the validation of GLEAM itself compared to the merged product and other individual ones).

**Response:**

Thank you for your very thoughtful comment. Firstly, ERA5, MERRA2 and GLDAS are based on climate models, while GLEAM is not. Secondly, GLDAS is driven by atmospheric observations, while GLEAM contains input data from land surface, such as soil moisture and vegetation optical depth. Therefore, GLEAM is the most independent in these ET data. We have added the validation of GLEAM itself compared to the merged product and other individual ones to the revised manuscript. Lines 345-377 read: "

[revised manuscript text omitted]

*Similar to Table 2, Table 3 shows the performance of ET products in different ecosystems on a monthly scale. Compared with daily scale, the performance of each product has changed, among which all of the R becomes higher. REA has higher R and lower RMSD than individual products at 23 DBF, 42 ENF, 8 SAV and 19 WET sites. It has an optimal R or RMSD at 17 CRO, 9 MF and 9 OSH sites. At other 64 sites, REA has a worse performance than at least one individual product. For 1 DNF site, REA has lower R and higher RMSD than ERA5. For 13 EBF sites, REA has lower R and higher RMSD than GLEAM and ERA5. For 34 GRA sites, ERA5 has a lower R of 0.77 than MERRA2 and GLEAM, and a higher RMSD of 27.53 mm per month than GLEAM, ERA5 and GLDAS. For 6 WSA sites, REA has a lower R of 0.73 than GLEAM, and a higher RMSD of 34.49 mm per day than GLEAM, GLDAS and ERA5. Similar to the daily scale, REA does not have a better performance than any individual product in all ecosystems, but is superior to at least one individual one. Similarly, the Taylor charts of monthly Ground-measured ET and ET from the different products in 11 ecosystems are put in support information (Fig. S3).".*

L120: Monthly data ... for what purpose is this monthly data used?

**Response:**

Thank you for your very thoughtful comment. Previous studies show that there is a close relationship between the quality of land evaporation and vegetation (Miralles et al., 2016). Monthly GIMMS NDVI3g data is used to study how the quality of these land evaporation data sets change with vegetation, according to the correlation coefficients between multiple data sets and station-observed data under different vegetation conditions. We have added the description of the purpose of using this data. Lines 122-125 read: "*Monthly GIMMS NDVI3g data with a spatial resolution of 0.25° from the Global Inventory Modeling and Mapping Studies (GIMMS) was used to study how the quality of land evaporation data sets change with vegetation in our study (Pinzon & Tucker 2014), with the time span from 1982 to 2014, which is available from http://ecocast.arc.nasa.gov/data/pub/gimms/3g/.*".

L227: Is GLDAS a reanalysis? If yes, okay.

**Response:**

Thank you for pointing this out. GLDAS is a model-based product. We have changed "reanalysis" to "model-based" in the revised manuscript. Lines 229-230 read:

*"Reliability Ensemble Averaging (REA) method (Giorgi and Mearns, 2002; Xu et al., 2010) was used to combine multiple sets of model-based ET data into a single product."*.